# FOAM: Frequency and Operator Error-Based Adaptive Damping Method for Reducing Staleness-Oriented Error for Shampoo

**Kyunghun Nam** [1]  **Sumyeong Ahn** [1]

## Abstract

Shampoo is attracting considerable attention for its superior performance on large-scale optimization benchmarks; yet it faces a significant practical bottleneck: the prohibitive computational overhead of matrix inversion. To mitigate this, practitioners typically rely on *stale* preconditioner updates, creating a fundamental trade-off between computational efficiency and optimization fidelity. In this work, we provide a theoretical study of staleness through the complementary lenses of convergence and stability. While staleness improves computational efficiency, it inherently degrades performance and introduces numerical instability. Crucially, we identify that damping, acting as a numerical stabilizer, can effectively suppress these negative effects. Guided by this analysis, we propose FOAM, an adaptive algorithm that stabilizes training by dynamically controlling both the damping factor and the eigendecomposition frequency based on an approximation of the staleness-oriented error. Experimental results demonstrate that FOAM reduces wall-clock time compared to standard Shampoo while maintaining robust convergence.

## 1. Introduction

The advancement of deep learning has been propelled by massive scaling in both model architectures and datasets (Kaplan et al., 2020; Hoffmann et al., 2022; Achiam et al., 2023). From an optimization standpoint, classical first-order methods—such as SGD (Robbins & Monro, 1951) and Momentum (Polyak, 1964)—remain the default choice thanks to their low per-iteration overhead; yet, they exhibit slow convergence rates (Bottou et al., 2018). In partic-

ular, when the loss landscape is ill-conditioned (*i.e.,* exhibits highly non-uniform curvature across parameter dimensions) (Li et al., 2018; Yao et al., 2020), purely first-order updates may require a large number of iterations to make meaningful progress (Garrigos & Gower, 2023). This limitation has renewed interest in optimization approaches that incorporate richer curvature information while remaining computationally efficient, aiming to reduce wall-clock training time.

To address these geometric challenges, preconditioned gradient methods (*e.g.,* leveraging Hessian (Wall, 1948), Fisher information (Amari, 2016), Gauss-Newton method (Buffelli et al., 2024), second-moment estimator (Duchi et al., 2011)) have emerged as a promising solution. By employing a preconditioner matrix to rescale the gradient based on local curvature, this method effectively normalizes the loss landscape, thereby maximizing progress at each update step. While widely adopted for their their computational efficiency, these method typically rely on diagonal approximations to scale updates element-wise, thereby overlooking potential parameter correlations.

To overcome the limits of diagonal approximations, structured preconditioning–such as Kronecker product-based (Martens & Grosse, 2015; Gupta et al., 2018) or layerwise adaptivity (You et al., 2020)– has been proposed to capture these missing correlations efficiently. Notably, Shampoo (Gupta et al., 2018; Shi et al., 2023) captures within-layer parameter correlations more efficiently via Kronecker-factored statistics, thereby aciheving the superior performance demonstrated in recent benchmarks (Dahl et al., 2023; Kasimbeg et al., 2025).

However, Shampoo fundamentally requires computing the inverse $p$-th roots of the Kronecker factors. It is well-known that such inverse operations incur high computational costs. To address computational overhead, practitioners relied on a heuristic strategy—referred to as *stale* Shampoo—in which those expensive operations are performed at a fixed frequency. Although practically effective, this frequency choice remains a heuristic without a theoretical justification.

In this work, we show that staleness (*i.e.,* the delay in updating preconditioners) is not merely a convergence speed-

[1]KENTECH, Naju, Republic of Korea. Correspondence to: Sumyeong Ahn <sumyeongahn.kentech.ac.kr>.

*Proceedings of the 43rd International Conference on Machine Learning*, Seoul, South Korea. PMLR 306, 2026. Copyright 2026 by the author(s).

related issue but a fundamental threat to training stability. Specifically, through analyses of convergence and stability, we observe that staleness-induced errors simultaneously hinder convergence speed and severely degrade numerical robustness.

Building on these two observations, we find that damping — acting as a critical numerical stabilizer — must be meticulously calibrated to counteract staleness-induced errors. If the damping is insufficient relative to the measured operator gap, these errors can be amplified to the point of catastrophic training divergence. To mitigate this, we propose an adaptive algorithm that ensures robust stability by dynamically modulating both the damping factor and the update frequency in response to real-time error estimates.

**Contributions.** The main contributions are as follows:

- We establish that while staleness provides essential computational efficiency, it concurrently undermines both convergence and numerical stability. Furthermore, we theoretically demonstrate that damping — which acts as a numerical stabilizer — is an effective inhibitor of these adverse effects. To the best of our knowledge, this work is the first to use damping to counteract the inherent negative effects of staleness in optimization.

- Drawing on matrix perturbation theory, we propose an algorithm, FOAM[1], that dynamically regulates the damping factor to enhance numerical stability. The core mechanism adaptively increases damping to counteract errors as staleness increases; however, guided by our analysis of the negative impact of excessive damping, it precisely modulates these values to maintain an appropriate balance between error suppression and the preservation of essential preconditioning information—a common side effect of an excessively high damping factor.

- We validate our framework through comprehensive experiments on large-scale benchmarks, in which our adaptive mechanism consistently yields superior performance and accelerates convergence compared with stale Shampoo. Furthermore, we verify the algorithm's robustness across a wide range of hyperparameter settings. Through in-depth analyses, we demonstrate that our experimental results substantiate our theoretical assertions, bridging the gap between our analysis and practical optimization.

## 2. Related Work

**Preconditioned gradient methods.** Preconditioned gradient methods can be viewed as gradient descent on reparameterized variables (Grosse, 2022). Classical approaches

(*e.g.,* Newton (Nesterov et al., 2018) and natural gradient (Amari, 2016)) leverage the Hessian or Fisher information, while widely used adaptive methods such as Adagrad/Adam (Duchi et al., 2011; Kingma, 2014) typically approximate (empirical) curvature using diagonal statistics (Kunstner et al., 2019). Shampoo (Gupta et al., 2018; Shi et al., 2023) advances this line by employing Kronecker-factored, block-structured second-moment matrices, capturing within-layer correlations.

**Shampoo family.** A growing family of methods builds on Shampoo's Kronecker structure, including CASPR (Duvvuri et al., 2024), SOAP (Vyas et al., 2025), Muon (Jordan et al., 2024), SPlus (Frans et al., 2026), and KL-Shampoo (Lin et al., 2026). CASPR improves approximation quality via Kronecker-sum combinations of axis preconditioners (Duvvuri et al., 2024), but still requires expensive inverse(-root) computations. SOAP and related implementations often reuse stale eigenspaces and periodically refresh them via iterative schemes (*e.g.,* QR) (Vyas et al., 2025); this implicitly assumes eigenspace stability, which can break when eigengaps or damping are small. Moreover, refresh triggers based on diagonalization residuals (Eschenhagen et al., 2026) need not track the inverse-root *operator* error that actually governs the update and can therefore be inefficient or miscalibrated. SPlus (Frans et al., 2026) targets instabilities from stale reuse with practical stabilizers.

**Theoretical analysis and hyperparameters.** Recent theory formalizes Shampoo-type updates through non-Euclidean steepest descent / proximal and LMO viewpoints (Bernstein & Newhouse, 2024; Pethick et al., 2025), and studies structured-preconditioner regret bounds (Xie et al., 2025). The inverse-root exponent $p$ is another key choice as explained by the Kronecker approximation quality (Morwani et al., 2025). Damping is routinely introduced for stable inverse computations (Moré, 2006; Martens & Grosse, 2015; Ishikawa & Karakida, 2024), but has received less direct focus than step size (Anil et al., 2020; Agarwal et al., 2020; Zhang et al., 2025) or batch size (Ishikawa & Karakida, 2024; Kim & hwan Oh, 2025; Marek et al., 2026). Lastly, Damaskinos et al. (2018) dampens **stale gradients** in asynchronous SGD via a scalar weight, whereas our work controls **stale preconditioners** in Shampoo by adaptively tuning the inverse-root damping $\epsilon$.

## 3. Preliminary

This section formalizes the notation and stale Shampoo. For a more detailed background, we refer to Appendix B.2.

### 3.1. Notation and Shampoo Optimizer

In this section, we establish the mathematical notation for this study and introduce the Shampoo optimizer, along with

---

[1]We provide our official code in https://github.com/REAL-KENTECH/FOAM.git

---

**Algorithm 1** Generalized Shampoo with Modular $\mathcal{U}(\cdot)$

---

**Require:** Learning rate $\eta$; Momentum coefficient $\beta \in (0,1)$; Total steps $T$; Check period $\mathbf{f}$; Update criteria $\mathcal{U}(\cdot)$; inverse-root power $p$; Damping parameter $\epsilon_0$;

1: **Initialize:** $W_1, L_0^{-1/p}, R_0^{-1/p}, L_0 = \mathbf{0}_m, R_0 = \mathbf{0}_n$
2: **for** $t = 1$ **to** $T$ **do**
3:     Receive a loss function $f_t(\cdot) : \mathbb{R}^{m \times n} \to \mathbb{R}$
4:     $G_t = \nabla f_t(W_t)$
5:     $L_t = \beta L_{t-1} + (1-\beta) G_t G_t^\top$
6:     $R_t = \beta R_{t-1} + (1-\beta) G_t^\top G_t$
7:     /* Update rule: Stale or **FOAM** */
8:     $\boxed{(\hat{L}_{t+1}^{-1/p}, \hat{R}_{t+1}^{-1/p}, \epsilon_{t+1}) = \mathcal{U}(L_t, R_t, \hat{L}_t^{-1/p}, \hat{R}_t^{-1/p}, \epsilon_t, t)}$
9:     /* Model parameter update */
10:     $W_{t+1} = W_t - \eta \, \hat{L}_t^{-1/p} G_t \, \hat{R}_t^{-1/p}$
11: **end for**

---

**Algorithm 2** Stale Update for Shampoo

---

**Require:** Current factors $L_t, R_t$; Inverse-root factors $\hat{L}_t^{-1/p}, \hat{L}_t^{-1/p}$; Current iteration $t$
**Hyperparameter:** Stale frequency $\mathbf{f}$; Base damping $\epsilon_0$

1: **if** $((t-1)\%\mathbf{f}) = 0$ **then**
2:     /* Run **EVD** and update $\hat{L}_t^{-1/p}, \hat{R}_t^{-1/p}$ */
3:     Compute $Q_L D_L Q_L^\top = \text{EVD}(L_t)$
4:     Compute $Q_R D_R Q_R^\top = \text{EVD}(R_t)$
5:     $\hat{L}_{t+1}^{-1/p} \leftarrow Q_L(D_L + \epsilon_0 \mathbf{I}_m)^{-1/p} Q_L^\top$
6:     $\hat{R}_{t+1}^{-1/p} \leftarrow Q_R(D_R + \epsilon_0 \mathbf{I}_n)^{-1/p} Q_R^\top$
7: **else**
8:     $\hat{L}_{t+1}^{-1/p} \leftarrow \hat{L}_t^{-1/p}, \hat{R}_{t+1}^{-1/p} \leftarrow \hat{R}_t^{-1/p}$     // No update
9: **end if**
10: **return** $(\hat{L}_{t+1}^{-1/p}, \hat{R}_{t+1}^{-1/p}, \text{-})$

---

its stale update mechanism.

**Notations.** The vector space of $m \times n$ is denoted by $\mathbb{R}^{m \times n}$; $\mathbf{0}_m \in \mathbb{R}^{m \times m}$ is the zero matrix, and the $m \times m$ identity matrix is denoted by $\mathbf{I}_m$. For $A \in \mathbb{R}^{m \times n}$, $\|A\|_F$ and $\|A\|_2$ represent its Frobenius and spectral norms, respectively. For a symmetric matrix $A$, we denote its eigenvalues in descending order by $\{\lambda_i(A)\}_{i=1}^n$. The Kronecker product of $A$ and $B$ is denoted by $A \otimes B$. We use the Loewner order $A \succeq B$ to indicate that $A - B$ is positive semidefinite. For a symmetric positive definite matrix $A$ and $\alpha \in \mathbb{R}$, the matrix power is defined via eigendecomposition (EVD(A)) as $A^\alpha := PD^\alpha P^\top$, where $A = PDP^\top$.

**Shampoo.** Shampoo optimizer[2], presented as in Algorithm 1, provides a preconditioning matrix-valued parameter

---

[2]Depending on the choice of the root power $p$, this framework covers both the original Shampoo for $p = 4$ (Gupta et al., 2018) and Shampoo[2] for $p = 2$ (Morwani et al., 2025).

$W_t \in \mathbb{R}^{m \times n}$. It maintains the Kronecker-factored second-moment statistics ($L_t$ and $R_t$) via exponential moving-average updates of the gradients with parameter $\beta$. This code is written to be a flexible execution loop, with the specific preconditioning strategy encapsulated in an interchangeable $\mathcal{U}(\cdot)$ procedure highlighted in the blue box. The output of $\mathcal{U}(\cdot)$ consists of inverse-root factors applied to the gradient $G_t$ for the parameter update. Via the Kronecker identity, the update becomes $w_{t+1} = w_t - \eta(\hat{L}_t^{-1/p} \otimes \hat{R}_t^{-1/p})g_t$, where $w_t := \text{vec}(W_t)$ and $g_t := \text{vec}(G_t)$.

**Stale update.** As a baseline instance of the generalized framework described in Algorithm 1, the Stale update follows a fixed frequency schedule without any auxiliary execution conditions (please see Algorithm 2). Under this configuration, the update routine is invoked unconditionally every $\mathbf{f}$ steps to perform a full eigendecomposition (EVD) for both $L_t$ and $R_t$. Following the EVD, a fixed base damping $\epsilon_0$ is applied to construct the inverse root factors $\hat{L}_t^{-1/p}$ and $\hat{R}_t^{-1/p}$. These factors are then returned to the main skeleton for periodic reuse.

# 4. Theoretical Objectives: Stability and Regret

We introduce the central quantities we analyze and control: the staleness-induced operator gaps and discounted regret.

## 4.1. Objective 1: Staleness-induced Preconditioner Gap

This section formalizes the concept of the preconditioner gap, defined as the discrepancy $P_t - \hat{P}_t$ between the ideal fresh preconditioner and its stale counterpart. From this error, we derive the operator gaps, $\Delta_L$ and $\Delta_R$, which represent the operator gap within each Kronecker inverse-root factor. We demonstrate that regulating these factors is not only a sufficient condition for bounding the total gap but also a more computationally efficient strategy for maintaining stability.

**Preconditioner gap.** The numerical stability of the stale update is fundamentally governed by the preconditioner gap. Let $\mathbf{f} \in \mathbb{N}$ denote the refresh period and $t_0(t) := \max\{s \leq t : s \equiv 0 \pmod{\mathbf{f}}\}$ the most recent refresh time. We define the fresh and stale preconditioners as:

$$P_t := (L_t + \epsilon_0 \mathbf{I}_m)^{-1/p} \otimes (R_t + \epsilon_0 \mathbf{I}_n)^{-1/p},$$
$$\hat{P}_t := \hat{L}_t^{-1/p} \otimes \hat{R}_t^{-1/p},$$

where $\hat{L}_t := L_{t_0(t)}$ and $(\hat{L}_t)^{-1/p} := (L_{t_0(t)} + \epsilon_0 \mathbf{I}_m)^{-1/p}$ are the stale Kronecker factors. We call $L_t - \hat{L}_t$ as drift. The same applies to $\hat{R}_t, (\hat{R}_t)^{-1/p}$.

**Preconditioner gap decomposition.** To analyze how staleness affects the update $w_{t+1} = w_t - \eta \hat{P}_t g_t$, we decompose the preconditioner gap into its constituent components using

Kronecker product identities:

$$P_t - \hat{P}_t = \Delta_R(t) \otimes \hat{L}_t^{-1/p} + \hat{R}_t^{-1/p} \otimes \Delta_L(t) + \Delta_R(t) \otimes \Delta_L(t)..$$

A detailed derivation of this decomposition is provided in Appendix D. In this expression, we define the individual operator errors for each factor as:

$$\Delta_L(t) := (L_t + \epsilon_0 \mathbf{I}_m)^{-1/p} - \hat{L}_t^{-1/p},$$
$$\Delta_R(t) := (R_t + \epsilon_0 \mathbf{I}_n)^{-1/p} - \hat{R}_t^{-1/p}.$$

**Why focus on individual operator gaps?** Focusing on individual operator gaps $\Delta_L$ and $\Delta_R$ is a strategic choice that balances theoretical rigor with practical efficiency. First, by the triangle inequality and the multiplicative property of the norm, the magnitude of the preconditioner gap is directly upper-bounded by these individual factors; thus, bounding $\Delta$ is mathematically sufficient to stabilize the entire system. Second, from a computational standpoint, explicitly forming or monitoring the full $mn \times mn$ preconditioner is prohibitively expensive, scaling quadratically with the number of parameters. By regulating only the Kronecker factors ($m \times m$ and $n \times n$), we avoid the inefficiency of the full Kronecker product, thereby maintaining optimization stability with minimal overhead.

### 4.2. Objective 2: Discounted Regret

As the second core metric, we analyze the discounted regret to evaluate convergence performance under temporal staleness. The definition of the discounted regret is:

$$\sum_{t=1}^{T} \beta^{T-t} f_t(W_t) - \sum_{t=1}^{T} \beta^{T-t} f_t(W^\star),$$

where $\beta \in (0,1)$ denotes the discount factor and $W^\star$ is the comparator. Unlike standard regret, this formulation prioritizes more recent performance, which is essential for capturing the dynamic nature of an optimizer that reuses stale statistics.

## 5. Staleness Error and Its Inhibitor $\epsilon$

In this section, we derive mathematical upper bounds for our dual objectives and analyze their physical implications. We show that increasing the refresh period $\mathbf{f}$ amplifies the staleness error, which we categorize into two distinct forms:

- **Operator-gap staleness error**: Quantifies the operator gap $\Delta$ that undermines preconditioning stability.

- **Regret staleness error**: Measures the cumulative impact of stale statistics on convergence performance.

While the damping parameter $\epsilon$ serves as a primary inhibitor of these errors, it systematically suppresses the instability

induced by delayed updates (Section 5.1 and 5.2). However, there is a critical trade-off: while a larger $\epsilon$ effectively blocks the staleness error, an excessively high value can also hide the useful information needed for fast training, thereby reducing learning efficiency (Section 5.2).

### 5.1. Operator-gap Staleness Error

**Stale statistics.** The first step in our analysis is to quantify the optimizer's operator gap. This is measured by the discrepancy between the true preconditioner and its stale versions. We define the filtration $\mathcal{F}_t := \sigma(W_1, G_1, \ldots, G_t)$, where $W_1$ is the initialization. Our analysis relies on the following assumption about the stochastic gradient noise.

**Assumption 5.1.** For some constants $R > 0$ and $\sigma > 0$, the conditional mean satisfies $\|\mathbb{E}[G_t \mid \mathcal{F}_{t-1}]\|_2 \leq R$, and the centered noise is conditionally sub-Gaussian in spectral norm: for all $u \geq 0$ and all $1 \leq t \leq T$:

$$\mathbb{P}\Big(\big\|G_t - \mathbb{E}[G_t \mid \mathcal{F}_{t-1}]\big\|_2 \geq u \,\Big|\, \mathcal{F}_{t-1}\Big) \leq 2\exp\left(-\frac{u^2}{2\sigma^2}\right).$$

Under these conditions, we derive a high-probability bound on the operator gap, *i.e.*, $\|\Delta_L(t)\|$ and $\|\Delta_R(t)\|$, making the dependence on the refresh period $\mathbf{f}$ and damping $\epsilon$ explicit. By analogous arguments for the right factor, the full analysis for $\Delta_R(t)$ is deferred to Appendix E. Henceforth, we focus on the left factor for brevity.

**Lemma 5.2** (Operator Gap Bound). *Under Assumption 5.1, for any $\epsilon_0 > 0$ and any $p \geq 1$, with probability at least $1 - \delta$, the following holds simultaneously for all $1 \leq t \leq T$:*

$$\|\Delta_L(t)\|_F \leq \frac{1}{p\,\epsilon_0^{(p+1)/p}} \left\|L_t - \hat{L}_t\right\|_F$$

$$\leq \underbrace{\frac{2\sqrt{m}}{p\,\epsilon_0^{(p+1)/p}}(1 - \beta^{\mathbf{f}})\,R_{\mathrm{SG}}^2}_{\textit{Operator-gap staleness error}}, \qquad (1)$$

*where $R_{\mathrm{SG}} := R + K\sigma\sqrt{\log(T/\delta)}$ for a universal constant $K > 0$ and $0 < \delta < 1$.*

Refer to Appendix E for the proof. The bound in (1) highlights two distinct mechanisms that drive operator error.

**Lipschitz instability (inverse-root sensitivity).** The first inequality isolates the sensitivity of the inverse-root map through the factor $1/(p\epsilon_0^{(p+1)/p})$ (cf. Lemma C.1). This term is an upper bound on the Lipschitz constant; as $\epsilon \to 0$, the map becomes hypersensitive, so even small drift can induce a large operator gap. Thus, $\epsilon$ is not merely a numerical stabilizer but also a regulator of operator stability under staleness. We also analyze spectral stability under staleness, including the eigenvalue gap and eigenspace rotation in Appendix F. Importantly, both effects inherit the same $\epsilon$-dependent sensitivity of the inverse-root map.

**EMA evolution across a stale window.** The second inequality quantifies how stale statistics lag behind fresh ones over a window of length $\mathbf{f}$, as captured by $(1 - \beta^{\mathbf{f}})$. When $\beta$ is close to 1, $(1 - \beta^{\mathbf{f}}) \approx \mathbf{f}(1 - \beta)$, skipping refreshes yields a nearly linear growth of drift with $\mathbf{f}$. This is the predictable component of operator-gap staleness error.

**$\epsilon$ as a staleness error inhibitor.** Together, these effects reveal a fundamental relationship: while increasing $\mathbf{f}$ reduces computational overhead, it inherently amplifies the staleness error. Within this framework, $\epsilon$ serves as a critical inhibitor, mitigating the sensitivity of the inverse-root factors to ensure stable operator gap bounds. This interaction constitutes the local manifestation of the staleness error.

### 5.2. Regret Staleness Error

To determine the effect of operator error on convergence, we derive a discounted regret bound for stale Shampoo. This bound allows us to isolate the regret staleness error, which is the specific convergence penalty caused by the refresh interval $\mathbf{f}$. Furthermore, we demonstrate that an excessively large inhibitor (*i.e.,* large $\epsilon_0$) leads to adverse side effects, identifying a fundamental limit to the preconditioning benefit. For results for stale Shampoo with $p = 4$ and the full proof, see Appendix G.

**Assumption 5.3.** The loss $f_t(\cdot)$ is convex for all $1 \leq t \leq T$.

**Theorem 5.4** (Discounted Regret Bound). *Assume Assumption 5.1 and Assumption 5.3. Furthermore, assume* $\mathrm{rank}(G_t) \leq r \leq \min(m, n)$ *and* $\max_{t \in T} \|W_t - W^\star\|_F = D(W^\star) =: D$ *for all* $W^\star \in \mathbb{R}^{m \times n}$. *Then, with probability at least* $1 - \delta$, *the discounted regret of stale Shampoo with* $p = 2$ *satisfies*

$$
\sum_{t=1}^{T} \beta^{T-t} f_t(W_t) - \sum_{t=1}^{T} \beta^{T-t} f_t(W^\star)
$$
$$
\leq \frac{1}{2\eta} \left\{ \epsilon_0 D^2 + \frac{\sqrt{R_{\mathrm{SG}}^2 + \epsilon_0}}{\sqrt{\epsilon_0}} D^2 R_{\mathrm{SG}}^2 + \frac{D^2(R_{\mathrm{SG}}^2 + \epsilon_0)}{\beta} \right\}
$$
$$
+ \underbrace{\frac{\eta}{1-\beta} \left\{ \frac{r(1 - \beta^{\mathbf{f}}) R_{\mathrm{SG}}^4}{\epsilon_0^2} \right\}}_{\text{Regret staleness error}} \tag{2}
$$
$$
+ \frac{\eta r m n}{2(1-\beta)} \left\{ \log\left( \epsilon_0 + \frac{(1 - \beta^T) r R_{\mathrm{SG}}^2}{mn} \right) - \log(\epsilon) \right\},
$$

*where* $R_{\mathrm{SG}} = R + K\sigma\sqrt{\log(T/\delta)}$ *and* $K > 0$ *is a universal constant.*

**Validation of the stability logic.** The regret staleness error term in (2) scales linearly with the drift factor $(1 - \beta^{\mathbf{f}})$ and inversely with $\epsilon_0^2$. Thus, as the refresh period $\mathbf{f}$ increases to reduce computational overhead, $\epsilon_0$ must increase to prevent staleness-driven error accumulation.

**The risk of over-inhibition.** The regret bound also reveals a countervailing constraint not visible at the operator level: the leading term includes $\epsilon_0 D^2$, reflecting the base cost of damping. Hence, overly large $\epsilon_0$ inflates the regret through the distance component, producing an intrinsic ceiling on admissible damping. Overall, stable and efficient stale Shampoo requires controlling the operator gap while tuning $\epsilon$ to balance inverse-root stability against preconditioning strength, rather than fixing it a priori.

## 6. Proposed Method: FOAM Update

To address the staleness error —as established in Section 5—we propose the **F**requency and **O**perator-error based **A**daptive **D**amping **M**ethod (FOAM). The key idea is to treat the spectral refresh frequency and the damping level $\epsilon$ as *coupled* control variables. In the following, we detail the procedure of FOAM.

---

**Algorithm 3** FOAM factor-wise update rule at $t$ ($L$ factor)

---

**Require:** $L_t$; $\mathcal{S}_{t-1}^L = (Q_L, D_L)$; $\hat{L}_{t-1}^{-1/p}$; $\epsilon_{t-1}^L$; $t$
**Hyperparameter:** Check period $\mathbf{f}$; max damping $\epsilon_{\max}$;
    base damping $\epsilon_0 \in (0, \epsilon_{\max})$; threshold $\tau \in (0, 1)$
1: **if** $t = 1$ **then**
2:     Compute $Q_L D_L Q_L^\top = \mathrm{EVD}(L_t)$
3:     $\epsilon_t^L \leftarrow \epsilon_0$
4:     $\hat{L}_t^{-1/p} \leftarrow Q_L(D_L + \epsilon_t^L \mathbf{I}_m)^{-1/p} Q_L^\top$
5:     $\mathcal{S}_t^L \leftarrow (Q_L, D_L)$
6: **else if** $(t - 1) \bmod \mathbf{f} = 0$ **then**
7:     /* 1. Compute error proxy */
8:     Compute $h_t^L = \mathrm{RC}_t^L(\epsilon_{t-1}^L) \cdot \alpha_t^L(\epsilon_{t-1}^L)/p$
9:     /* 2. Update damping */
10:     $\tilde{\epsilon}_t^L \leftarrow \max\left( \epsilon_0, \epsilon_{t-1}^L \frac{h_t^L}{\tau} \right)$
11:     **if** $\tilde{\epsilon}_t^L \leq \epsilon_{\max}$ **then**
12:         /* No EVD: Only apply updated damping */
13:         $\epsilon_t^L \leftarrow \tilde{\epsilon}_t^L$
14:         $\hat{L}_t^{-1/p} \leftarrow Q_L(D_L + \epsilon_t^L \mathbf{I}_m)^{-1/p} Q_L^\top$
15:         $\mathcal{S}_t^L \leftarrow \mathcal{S}_{t-1}^L$
16:     **else**
17:         /* Refresh by EVD and reset damping */
18:         Compute $Q_L D_L Q_L^\top = \mathrm{EVD}(L_t)$
19:         $\epsilon_t^L \leftarrow \epsilon_0$
20:         $\hat{L}_t^{-1/p} \leftarrow Q_L(D_L + \epsilon_t^L \mathbf{I}_m)^{-1/p} Q_L^\top$
21:         $\mathcal{S}_t^L \leftarrow (Q_L, D_L)$
22:     **end if**
23: **else**
24:     $\hat{L}_t^{-1/p} \leftarrow \hat{L}_{t-1}^{-1/p}$
25:     $\epsilon_t^L \leftarrow \epsilon_{t-1}^L$
26:     $\mathcal{S}_t^L \leftarrow \mathcal{S}_{t-1}^L$
27: **end if**
28: **return** $(\hat{L}_t^{-1/p}, \mathcal{S}_t^L, \epsilon_t^L)$

---

### 6.1. Update Inverse-root Factors for Shampoo via FOAM

The objective of FOAM is to reduce the number of costly eigendecompositions while maintaining a bounded oper-

ator gap. It does so via a feedback control loop over $\epsilon$, using an inexpensive proxy that can be evaluated in the stale eigenspaces.

The update mechanism consists of three phases, summarized in Algorithm 3. For brevity, we state the update only for the left Kronecker factor $L_t$. The right factor $R_t$ is updated by the same rule with the substitutions $L \mapsto R$, $m \mapsto n$, $Q_L \mapsto Q_R$, $D_L \mapsto D_R$, and $\epsilon_t^L \mapsto \epsilon_t^R$. For the remainder of Section 6, we use one-factor notation $h_t, \epsilon_t, \hat{L}_t$ for brevity; the right factor follows by the corresponding substitutions. Within the overall Shampoo procedure in Algorithm 1,3 defines the update criterion (indicated in the blue box, $\mathcal{U}(\cdot)$) that governs both damping adaptation and refresh decisions.

**Phase 1: Sensing (operator gap estimation).** This phase estimates whether the current damping level $\epsilon_{t-1}$ is sufficient to suppress the operator gap induced by the reuse. Rather than computing a fresh eigendecomposition to evaluate the true inverse-root operator gap, FOAM uses a proxy:

$$h_t = \frac{\alpha(\epsilon_{t-1})}{p} \, \text{RC}(\epsilon_{t-1}),$$

where $\text{RC}(\cdot)/p$ measures the mismatch between the current preconditioners and the stale ones, and $\alpha(\epsilon)$ is a factor that converts this mismatch into a tighter estimate of the operator gap. Detailed definitions and the theoretical derivation of these terms are provided in Section 6.2

**Phase 2: Adapting (dynamic damping).** Given the proxy $h_t$, FOAM enforces the stability condition derived in Section 5.1 via a multiplicative feedback rule:

$$\epsilon_t \;\leftarrow\; \max\left(\epsilon_0, \; \epsilon_{t-1} \cdot \frac{h_t}{\tau}\right).$$

If the estimated error $h_t$ exceeds a threshold $\tau$, we increase $\epsilon_t$ to stabilize the update. If $h_t \le \tau$, we decrease $\epsilon_t$ towards its baseline $\epsilon_0$. Effectively, $\epsilon_t$ serves as an inhibitor that suppresses staleness errors. This allows FOAM to reuse old statistics efficiently while keeping the total error within a safe, manageable range.

**Phase 3: Frequency control (triggered refresh).** While increasing $\epsilon$ helps stability, too much of it can reduce the benefits of the optimizer, as analyzed in Section 5.2. To prevent this, FOAM sets a maximum limit, $\epsilon_{\max}$. We continue to reuse old statistics as long as $\epsilon$ stays below this limit to save time. However, if the error becomes so large that $\epsilon$ exceeds $\epsilon_{\max}$, FOAM triggers a full refresh of the statistics and resets $\epsilon$ to its baseline. This design ensures we perform expensive updates only when damping is no longer sufficient to maintain stability.

### 6.2. Construction of the Error Proxy $h_t$

We now introduce the error proxy $h_t$, a tractable surrogate designed to estimate the operator gap $\Delta$ using only the avail-

able statistics. It serves as a real-time sensor that monitors operator error to determine exactly when $\epsilon$ must be adjusted to maintain stability. The design of $h_t$ is governed by three primary considerations: theoretical grounding in perturbation theory, actionable control logic for $\epsilon$, and computational tractability.

**Relative perturbation as a stability metric.** The decision to refresh a preconditioner should be based not on the absolute reuse error $\|\Delta_L(t)\|_F$, but on the relative changes it induces in the preconditioner- *i.e.*, $\|\Delta_L(t)\|_F$ normalized by the norm of the stale preconditioner currently in use. Drawing from matrix perturbation theory (Horn & Johnson, 2012; Bhatia, 2013), we define our sensing objective as controlling the relative inverse-root operator gap: $\frac{\|\Delta_L(t)\|_F}{\|(\hat{L}_t)^{-1/p}\|_F}$. Unlike standard diagonalization residuals (Eschenhagen et al., 2026), which mainly track how well an eigenspace diagonalizes the current statistic, the relative gap measures the fidelity of the inverse-root mapping itself. This mapping is anisotropic: it is far more sensitive to perturbations that act on (or mix into) low-spectrum directions, where small drifts can be strongly amplified by the inverse-root. By normalizing the magnitude of the damped operator in use, the relative operator gap yields a scale-free health signal that is more directly tied to the numerical stability of the Shampoo update (refer to Appendix I for a detailed comparison).

**Multiplicative damping via scaling logic.** We use the proxy $h_t$ as a feedback signal to update damping at check steps. The key point is that $h_t$ depends on $\epsilon$ through the whitening factors $(\hat{L}_t + \epsilon \mathbf{I}_m)^{-1/2}$: increasing $\epsilon$ downweights low-eigenvalue directions, precisely where inverse-root amplification is strongest, and therefore reduces the *relative* operator-error signal captured by $RC(\epsilon)$. This motivates a multiplicative feedback update (with a floor at $\epsilon_0$): if $h_t > \tau$ we increase $\epsilon$ to suppress sensitivity and restore stability, whereas if $h_t \le \tau$ we relax $\epsilon$ toward $\epsilon_0$ to preserve preconditioning strength.

**Derivation of the tractable relative operator error.** While the exact relative operator gap is the ideal metric, computing it precisely at every step would require an eigendecomposition, which defeats the purpose of our method. We therefore seek a proxy that (i) is computable in the stale eigenspaces and (ii) upper-bounds the relative change induced by staleness.

To this end, we approximate the relative operator error and analyze the resulting first-order term. we obtain a bound on the *relative* magnitude of the first-order approximation $\Delta_L^{(1)}$:

$$\frac{\|\Delta_L^{(1)}\|_F}{\|\hat{L}_t^{-1/p}\|_F} \;\le\; \frac{\alpha}{p} \, \left\| (\hat{L}_t)^{-1/2} \, (L_t - \hat{L}_t) \, (\hat{L}_t)^{-1/2} \right\|_F \quad (3)$$

where $\alpha(\epsilon) := \|\hat{L}_t^{-1/p}\|_2 / \|\hat{L}_t^{-1/p}\|_F$ and $\text{RC}(\epsilon_{t-1}) :=$

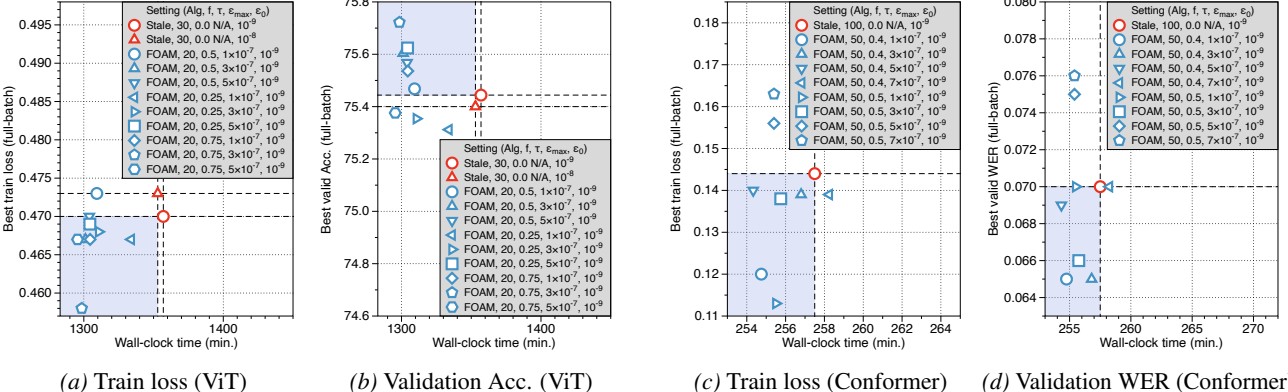

*(a)* Train loss (ViT)   *(b)* Validation Acc. (ViT)   *(c)* Train loss (Conformer)   *(d)* Validation WER (Conformer)

*Figure 1.* Wall-clock efficiency comparison between Shampoo with stale update and FOAM update. Figure 1a-1b presents the best training loss and validation accuracy for the ViT (ImageNet-1K) task, while Figure 1c-1d shows the training loss and WER for the Conformer (LibriSpeech) task. In all plots, the **blue rectangular areas** represent the region of superior performance, where FOAM achieves better convergence and final metrics in significantly less wall-clock time than the stale-update baselines.

$\|(\hat{L}_t)^{-1/2}(L_t - \hat{L}_t)(\hat{L}_t)^{-1/2}\|_F$. We call the RHS of (3) as $h_t$. A complete derivation of (3) is provided in Appendix H. We also present a synthetic experiment in Appendix K that shows that our proxy is a better approximation of the true relative error than the diagonalization error.

# 7. Experiments

In this section, we evaluate the performance of the proposed algorithm across several large-scale deep learning tasks.

## 7.1. Experimental Setup

This section details the experimental configurations and benchmarking protocols used to evaluate the wall-clock efficiency and robustness of FOAM diverse large-scale tasks.

**Tasks and benchmarks.** We evaluate the performance of FOAM on three distinct large-scale benchmarks representing different modalities: (1) image classification on ImageNet-1K (Deng et al., 2009) using a Vision Transformer (ViT-small) (Dosovitskiy et al., 2021), (2) automatic speech recognition (ASR) on LibriSpeech (Panayotov et al., 2015) using a Conformer architecture (Gulati et al., 2020) and (3) language modeling on Wikitext-103-v1 (Merity et al., 2016) using GPT-2 (Radford et al., 2019). These tasks exhibit diverse spectral characteristics, enabling us to assess the robustness of our adaptive sensing mechanism across varying gradient structures. For the result of GPT-2, refer to Appendix J.

**Implementation details.** Our implementation is integrated into the AlgoPerf (Kasimbeg et al., 2025) framework to provide a standardized, reproducible benchmarking environment. All models are trained for a fixed budget unless otherwise specified. For FOAM, we additionally sweep over the sensing frequency **f**, tolerance $\tau$, and maximum damp-

ing $\epsilon_{max}$, reporting the configuration that achieves the lowest full-batch loss.

**Evaluation metrics.** The primary metric for comparison is the total wall-clock time to run fixed epochs. For the vision task, we track training loss and top-1 validation accuracy. For the speech task, we monitor training loss and Word Error Rate (WER). For the language task, we monitor training loss and Perplexity (PPL). A method is considered superior if it achieves comparable or better final performance (accuracy/WER/PPL) while reducing total wall-clock time by decreasing the computational overhead of EVD.

**Experimental environment.** Experiments are conducted on high-performance GPU clusters to measure actual wall-clock throughput. For the ViT benchmarks, we use 4 A6000 GPUs, whereas the Conformer tasks are executed on 4 RTX Pro 6000 GPUs. Specific ablation studies are performed on a smaller scale using 2 RTX Pro 6000 GPUs. All software dependencies, including the CUDA version and framework specifications, are detailed in Appendix J.

## 7.2. Experimental Results

Across both ViT and Conformer benchmarks, FOAM exhibits a significant wall-clock advantage while matching or exceeding the final performance of Shampoo with a stale update, as illustrated in Figure 1. Specifically, for the ViT task, FOAM achieves lower training loss and higher validation accuracy in substantially less time than the stale baselines. A similar trend is observed in the Conformer experiments, where FOAM achieves a lower WER in less time. The blue rectangular regions in each plot highlight the area of superior performance in which FOAM resides.

Furthermore, these gains are consistent across a broad range of hyperparameters. A comprehensive sweep over

$(\mathbf{f}, \tau, \epsilon_{\max}, \epsilon_0)$ shows that the majority of tested configurations fall within the superior blue regions, demonstrating that the wall-clock advantage of FOAM is robust and not dependent on a single tuned setting. This stability confirms the effectiveness of the proxy-driven controller in maintaining performance across various training scenarios. Lastly, we refer to Appendix J for comparison experiments with other optimizers (*e.g.,* AdamW, SOAP).

### 7.3. Analysis

We provide a detailed examination of FOAM's internal dynamics, investigating the evolution of the adaptive damping parameter $\epsilon$, the relative reduction in eigendecomposition frequency for Kronecker factors, and the performance impact of its core components through systematic ablation studies on ViT. For these analysis, the hyperparameter configurations are set to $(\mathbf{f}, \tau, \epsilon_{\max}, \epsilon_0) = (20, 0.75, 3 \times 10^{-7}, 10^{-9})$ for ViT and $(50, 0.4, 1 \times 10^{-7}, 10^{-9})$ for Conformer. Other hyperparameters are referred to in Appendix J.

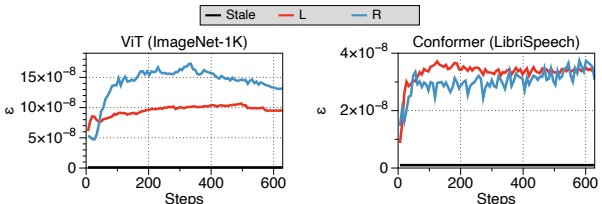

*Figure 2.* $\epsilon$ dynamics compared to the iterations.

**The evolution of adaptive damping $\epsilon$.** Figure 2 illustrates the adaptive $\epsilon$ dynamics of FOAM, contrasting it with the low, static damping maintained by the stale update baseline. While the stale baseline relies on a fixed, minimal value, FOAM adaptively raises $\epsilon_t$ as a compensatory mechanism to counteract spectral drift and ensure numerical robustness during the sensing phase. This proactive adjustment keeps the model stable even as the age of the Kronecker factors increases. Crucially, to prevent the side effect of over-damping — which can degrade preconditioning quality — the controller incorporates a modulation strategy that regulates the growth and relaxation of $\epsilon_t$. This feedback-driven mechanism ensures that damping remains within a safe and effective range, balancing error suppression with optimal convergence throughout training.

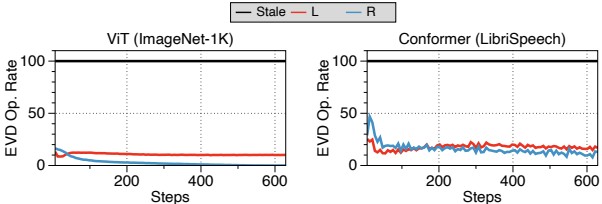

*Figure 3.* Relative rate of EVD function calls.

**EVD frequency.** Figure 3 quantifies the reduction in computational overhead by comparing the EVD frequency of

FOAM against the fixed-cadence stale update method, normalized to $100\%$. While the stale baseline performs EVD updates at every scheduled interval regardless of necessity, FOAM triggers refreshes only when the error proxy $h_t$ indicates significant spectral drift. In the ViT case, the number of EVD for $L$ and $R$ drops to approximately $10\%$ and $5\%$, respectively. A similar trend is observed in the Conformer benchmark. This selective refresh mechanism demonstrates that a large portion of scheduled updates in traditional stale preconditioning is computationally redundant, and by bypassing them, FOAM achieves direct wall-clock gains.

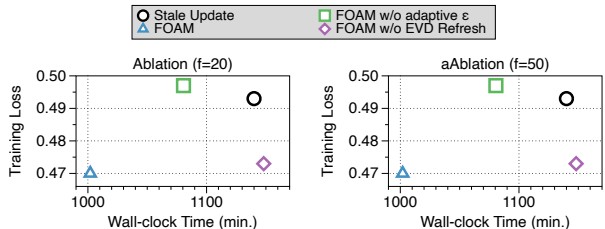

*Figure 4.* Ablation study of FOAM.

**Ablation study.** Figure 4 presents an ablation study on the ViT architecture, demonstrating the critical synergy between adaptive damping and selective EVD refreshes. When increasing the interval $\mathbf{f}$ to gain computational efficiency without utilizing adaptive $\epsilon$ compensation (green square), the training loss increases significantly as the stale preconditioner fails to account for spectral drift. Conversely, relying solely on $\epsilon$ control without triggering EVD refreshes (purple diamond) can reduce the training loss, but at the cost of significantly higher wall-clock time. Only the full configuration of FOAM (blue triangle) achieves the balance, simultaneously minimizing both training loss and computational overhead. These results confirm that integrating proxy-driven damping adaptation and well-timed basis refreshes is essential for maintaining high performance within a reduced wall-clock budget.

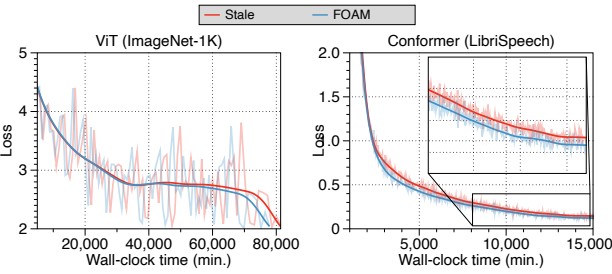

*Figure 5.* Learning curve. (Wall-clock time vs Loss)

**Learning curve.** As described in Figure 5, FOAM consistently achieves lower training loss compared to Stale across both the ViT and Conformer datasets. The magnified insets further reveal that FOAM maintains a superior convergence throughout training, confirming that its adaptive feedback

mechanism successfully preserves and stabilizes the optimization while operating at higher computational efficiency.

## 8. Conclusion

We presented FOAM, a mathematically grounded framework that resolves the computational bottleneck of preconditioned optimization by decoupling spectral sensing from costly refreshes. By introducing an error proxy and a feedback controller for adaptive damping, our method ensures stability while reducing the frequency of eigendecompositions—often by more than $80\%$ relative to a stale Shampoo baseline. Our extensive evaluation demonstrates that FOAM consistently achieves significant wall-clock speedups without sacrificing convergence quality or final performance across diverse modalities. Ultimately, FOAM bridges the gap between the superior optimization trajectories of Shampoo-style methods and the practical efficiency required for large-scale deep learning.

## Acknowledgements

This work was partly supported by Institute of Information & communications Technology Planning & Evaluation (IITP) grant funded by the Korea government(MSIT) (RS-2025-25464461, AI's Vision of Harmony: A Fair and Transparent Multimodal Agentic Platform for Conflict Mediation), and the Technology Innovation Program (RS-2025-02653102, An Unbreachable Multilayer AI-based Security Mechanism that Continuously Adapts and Evolves in Dynamic Conditions) funded by the Ministry of Trade Industry & Energy (MOTIE, Korea).

## Impact Statement

This paper presents FOAM, which aims to improve the computational efficiency and numerical robustness of preconditioned optimization in large-scale machine learning. By providing a theoretical foundation for managing curvature staleness through adaptive damping, this work aims to bridge the gap between high-fidelity optimization and practical training speed-ups. The broader impact of this research is twofold:

- Resource Efficiency: By accelerating convergence and reducing hyperparameter sensitivity, our approach contributes to the sustainable development of AI, potentially lowering the energy consumption and carbon footprint associated with training large models.

- Training Reliability: Enhancing the stability of the preconditioned method is crucial to developing more reliable and trustworthy AI systems, as it mitigates the risk of catastrophic divergence during resource-intensive training runs.

We believe that making sophisticated optimization techniques more accessible and stable is essential for the continued progress of the machine learning community.

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

# –Supplementary–
# `FOAM`: Frequency and Operator Error-Based Adaptive Damping Method for Reducing Staleness-Oriented Error for Shampoo

## Table of Contents

# A. Limitations and Future Works

While FOAM demonstrates robustness and efficiency, several avenues for improvement and extension remain:

- **Theoretical extensions to realistic settings:** Our current regret analysis focuses on convex settings. Extending this to non-convex (and possibly non-smooth) online learning regimes would provide guarantees that better reflect the dynamics of deep learning (Ahn et al., 2025; 2024; Cutkosky et al., 2023). Furthermore, our operator error bounds rely on sub-Gaussian assumptions; deriving analogous stability results under weaker conditions, such as bounded variance, represents a crucial step toward bridging the gap between theory and practice.

- **Reducing Hyperparameter complexity:** Experimentally, FOAM has consistently demonstrated superior efficiency and robustness compared to Stale Shampoo across a wide range of $\epsilon_{\max}$ and $\tau$. However, introducing two new hyperparameters expands the search space relative to the original Shampoo. Therefore, research aimed at minimizing the tuning cost for these parameters remains a critical direction for future work.

**Toward FOAM with SOAP-style (eigengap-aware motivation).** SOAP updates use only the eigenbases $Q_L, Q_R$ of Shampoo's Kronecker factors to rotate the gradients into an approximate eigenspace, apply an Adam step, and rotate back; the eigenvalues themselves do not explicitly enter the update. Consequently, when $Q_L, Q_R$ are refreshed only periodically (e.g., via QR-iteration), the dominant staleness effect is not merely a small "operator error" in the factors, but eigenspace drift: stale bases rotate gradients into wrong coordinates, so even a perfectly well-behaved diagonal update is applied along incorrect directions. Standard subspace-perturbation theory shows that the size of this drift is controlled by a ratio of the form "statistics drift/eigengap" (refer to Theorem F.2), so the same amount of change in the Kronecker factors can induce dramatically different basis errors depending on how well-separated the relevant eigenvalues are. This suggests that any Soap with FOAM style analysis should explicitly account for **eigengap-dependent** sensitivity when reasoning about (and bounding) the error induced by stale eigenbases updates.

# B. Notation and Preliminary

## B.1. Notation

The vector space of $m \times n$ matrices is denoted by $\mathbb{R}^{m \times n}$, and the $m \times m$ zero matrix and identity matrices are denoted by $\mathbf{0}_m$ and $\mathbf{I}_m$, respectively. We denote by $\mathbb{S}_n$ the set of symmetric $n \times n$ matrices and by $\mathbb{S}_{n,+}$ the cone of symmetric positive definite matrices. For $A \in \mathbb{R}^{m \times n}$, the trace is $\mathrm{Tr}(A)$. The inner product is $\langle A, B \rangle := \mathrm{Tr}(A^\top B)$, the Frobenius norm is $\|A\|_F := \sqrt{\langle A, A \rangle}$, and the spectral norm is $\|A\|_2 := \max_{\|x\|_2=1} \|Ax\|_2$. Given $H \in \mathbb{S}_{n,+}$, the energy norm and its dual are

$$\|x\|_H := \sqrt{x^\top H x}, \qquad \|x\|_H^\star := \sqrt{x^\top H^{-1} x}.$$

For $A \in \mathbb{S}_n$, we denote its eigenvalues in descending order by $\{\lambda_i(A)\}_{i=1}^n$ (or simply $\{\lambda_i\}_{i=1}^n$ when clear from context). The Kronecker product of $A \in \mathbb{R}^{m \times n}$ and $B \in \mathbb{R}^{p \times q}$ is denoted by $A \otimes B \in \mathbb{R}^{mp \times nq}$. For $A \in \mathbb{R}^{m \times n}$ with rows $a_1, \ldots, a_m$, we define $\mathrm{vec}(A) := (a_1 \, a_2 \, \cdots \, a_m)^\top \in \mathbb{R}^{mn}$. We write $A \succeq 0$ (resp. $A \succ 0$) to indicate that $A$ is positive semidefinite (resp. positive definite). The Loewner order $A \succeq B$ means $A - B \succeq 0$. Finally, for $A \in \mathbb{S}_{n,+}$ and $\alpha \in \mathbb{R}$, the matrix power is defined by $A^\alpha := P D^\alpha P^\top$, where $A = P D P^\top$ is an eigen-decomposition. If $A$ is merely PSD (not PD), $A^\alpha$ is well-defined for $\alpha \geq 0$. In the Appendix, we denote the base damping factor in Shampoo as $\epsilon > 0$.

## B.2. Preliminary

This subsection collects the background tools used in our analysis. We briefly review (i) the interpretation of preconditioning as reparameterization, (ii) discounted regret in Online Convex Optimization (OCO), and (iii) key results from matrix perturbation theory.

**Preconditioned gradient method** Preconditioned gradient methods can be viewed through reparameterization. Let us consider

$$\min_{w \in \mathbb{R}^m} \ell(w)$$
$$w = R\hat{w} \quad \Leftrightarrow \quad \hat{w} = R^{-1} w,$$

where $R \in \mathbb{R}^{n \times n}$ is invertible and $\hat{\ell}(\hat{w}) := \ell(R\hat{w})$ is defined. By the chain rule,

$$\nabla_{\hat{w}}\hat{\ell}(\hat{w}) = R^{\top}\nabla_w\ell(w).$$

A gradient step in the transformed space,

$$\hat{w}_{t+1} = \hat{w}_t - \eta\nabla_{\hat{w}}\hat{\ell}(\hat{w}_t),$$

corresponds in the original space to

$$w_{t+1} = R\hat{w}_{t+1} = w_t - \eta\underbrace{(RR^{\top})}_{=:A}\nabla_w\ell(w_t).$$

Thus, preconditioning by a matrix $A \succeq 0$ is equivalent to gradient descent after reparameterization via $R = A^{1/2}$.

**Online Convex Optimization**  We provide a concise overview of online convex optimization (OCO) and discounted regret; see Orabona (2019) for a comprehensive introduction. In OCO, for $T$ rounds, the learner selects $w_t \in \mathcal{W} \subseteq \mathbb{R}^n$, then a convex loss $f_t : \mathcal{W} \to \mathbb{R}$ is revealed, and the learner incurs a loss $f_t(w_t)$. The (static) regret against any comparator $w \in \mathcal{W}$ is

$$\text{Reg}_T(w) := \sum_{t=1}^{T} f_t(w_t) - \sum_{t=1}^{T} f_t(w).$$

**Discounted regret.**  To model nonstationarity and emphasize recent losses, several works consider *discounted regret* (Ahn et al., 2024; Jacobsen & Cutkosky, 2024; Zhang et al., 2024):

$$\text{DReg}_T(w) := \sum_{t=1}^{T} \beta^{T-t}f_t(w_t) - \sum_{t=1}^{T} \beta^{T-t}f_t(w),$$

where $\beta \in (0, 1]$ is a discount factor. Discounted regret is (i) a valuable surrogate for dynamic regret via discounted-to-dynamic conversion (Ahn et al., 2024) and (ii) a convenient metric for designing adaptive online algorithms in evolving environments (Zhang & Cutkosky, 2024).

**Why discounted regret here.**  Our stale strategy reuses preconditioner factors over a window, making the effective update rule time-varying and history-dependent. Following the discounted regret framework developed for Adam-type updates (Ahn et al., 2024), we derive discounted regret bounds for stale Shampoo with $p = 2$ and stale Shampoo with $p = 4$, explicitly capturing the dependence on the staleness frequency **f** and damping $\epsilon$.

**Matrix perturbation theory**  We briefly review perturbation tools for symmetric matrices to quantify how stale statistics deviate from fresh inverse-root factors and how these deviations propagate to the preconditioner. See Horn & Johnson (2012) for a detailed treatment.

Our analysis distinguishes between (i) the stability of eigenvalues and (ii) the sensitivity of invariant subspaces, and then connects these classical facts to inverse-root matrix functions.

- **Eigenvalue stability (Weyl's inequality; Lemma C.2).** Let $A \in \mathbb{S}_n$ and $E \in \mathbb{S}_n$, and denote by $\lambda_1(\cdot) \le \cdots \le \lambda_n(\cdot)$ the ordered eigenvalues. Weyl's inequality yields

$$\left|\lambda_i(A + E) - \lambda_i(A)\right| \le \|E\|_2 \qquad \text{for all } i \in \{1, \ldots, n\}.$$

  Thus, in the spectral norm, small perturbations imply uniformly small eigenvalue shifts.

- **Eigenspace sensitivity (Davis–Kahan; Lemma C.3).** In contrast, invariant subspaces can be much more sensitive. Let $V \in \mathbb{R}^{n \times d}$ be an orthonormal basis for an invariant subspace of $A$ associated with a target eigenvalue cluster/interval, and let $\hat{V}$ be the corresponding invariant subspace of $A + E$. If this cluster is separated from the rest of the spectrum by an eigengap $\delta > 0$, then Davis–Kahan gives a $\sin\Theta$ bound of the form

$$\|\sin\Theta(V, \hat{V})\|_2 \le \frac{\|E\|_2}{\delta},$$

  (see Lemma C.3 for the precise statement). Consequently, when eigenvalues are clustered (i.e., $\delta \downarrow 0$), even small perturbations can induce large rotations of the preconditioning basis, despite eigenvalues remaining stable.

- **Worst-case Lipschitz behavior of inverse-root maps and $\epsilon$-dependence.** Shampoo requires inverse-root matrix functions of the form

$$F_\epsilon(X) = (X + \epsilon \mathbf{I})^{-1/p}.$$

Over a regime where $X \succeq 0$ and $X + \epsilon \mathbf{I} \succ 0$, Lemma C.1 implies that $F_\epsilon$ is (globally) Lipschitz, with a worst-case constant controlled by the scalar map $f(t) = (t + \epsilon)^{-1/p}$ and its derivative

$$f'(t) = -\frac{1}{p}(t + \epsilon)^{-(p+1)/p}.$$

In particular, since $\sup_{t \geq 0} |f'(t)| = \frac{1}{p}\epsilon^{-(p+1)/p}$, one obtains the conservative worst-case scaling

$$\mathrm{Lip}(F_\epsilon) \;\lesssim\; \frac{1}{p\,\epsilon^{(p+1)/p}} \;=\; \mathcal{O}\!\left(\epsilon^{-(p+1)/p}\right).$$

We refer to this blow-up as *Lipschitz instability*: as $\epsilon \downarrow 0$, the worst-case upper bound on $\|F_\epsilon(X + E) - F_\epsilon(X)\|$ can become arbitrarily large, even for small damping factor $\|E\|$.

**Anisotropy and "small-eigenvalue directions."** Crucially, the above worst-case Lipschitz bound holds for *any* perturbation direction, but it becomes nearly tight only when the drift interacts strongly with the most sensitive low-spectrum directions (eigenvalues near $\epsilon$). This is inherently *anisotropic*: the first-order variation is governed by the Fréchet derivative $DF_\epsilon(X)[E]$, whose amplification factors depend on the spectrum of $X$ through $f'$ (and divided-difference coefficients in the non-commuting case).

To make the notion of "small-eigenvalue directions" explicit, consider the commuting (simultaneously diagonalizable) case: assume $XE = EX$ with $X, E \in \mathbb{S}_n$, so there exists an orthogonal $U$ such that

$$U^\top X U = \mathrm{diag}(\lambda_1, \ldots, \lambda_n), \qquad U^\top E U = \mathrm{diag}(\delta_1, \ldots, \delta_n).$$

Define the low-spectrum index set at level $\tau > 0$ by

$$\mathcal{I}_{\mathrm{low}}(\tau) := \{\, i : \lambda_i \leq \tau \,\}, \qquad P_{\mathrm{low}}(\tau) := \sum_{i \in \mathcal{I}_{\mathrm{low}}(\tau)} u_i u_i^\top,$$

where $u_i$ are the eigenvectors of $X$. Then "drift aligned with small-eigenvalue directions" can be quantified, e.g., by the concentration ratio

$$\rho_{\mathrm{low}}(E; \tau) := \frac{\|P_{\mathrm{low}}(\tau) E P_{\mathrm{low}}(\tau)\|_F}{\|E\|_F} \in [0, 1],$$

which equals $\left(\sum_{i \in \mathcal{I}_{\mathrm{low}}(\tau)} \delta_i^2\right)^{1/2} / \left(\sum_{i=1}^n \delta_i^2\right)^{1/2}$ in this commuting setting. When $\rho_{\mathrm{low}}(E; \tau) \approx 1$ with $\tau$ on the order of $\epsilon$, perturbations concentrate on the most sensitive low-spectrum subspace and the global worst-case Lipschitz bound is (approximately) attained; when $\rho_{\mathrm{low}}(E; \tau) \ll 1$, perturbations largely live in large-eigenvalue directions and the change in $(X + \epsilon \mathbf{I})^{-1/p}$ is typically much smaller than the worst-case prediction. This geometric viewpoint explains why a single global worst-case constant can be overly pessimistic in practice and motivates the direction-aware *a posteriori* error estimator developed in Section 6.2.

## C. Technical Lemmas

**Lemma C.1** (Lipschitz continuity of a matrix function $F(X) = X^{1/p}$). *Consider a compact domain $\mathcal{D} \subset \mathbb{S}_{n,+}$ defined as:*

$$\mathcal{D} = \{X \in \mathbb{S}_{n,+} \mid a\mathbf{I}_n \preceq X \preceq b\mathbf{I}_n\},$$

*where $0 < a \leq b < \infty$. For a given non-zero scalar $p$, the matrix function $F : \mathcal{D} \to \mathbb{S}_n$ defined by $F(X) = X^{1/p}$ is Lipschitz continuous with respect to the spectral norm $\|\cdot\|_2$. That is, for any $X, Y \in \mathcal{D}$, the following inequality holds:*

$$\|X^{1/p} - Y^{1/p}\|_2 \leq L\|X - Y\|_2,$$

*where the Lipschitz constant $L$ is given by the maximum magnitude of the derivative of the scalar function $f(t) = t^{1/p}$ on the interval $[a, b]$:*

$$L = \sup_{t \in [a,b]} \left| \frac{d}{dt} t^{1/p} \right| = \begin{cases} \frac{1}{p} a^{(1-p)/p} & \text{if } p \geq 1, \\ \frac{1}{p} b^{(1-p)/p} & \text{if } 0 < p < 1, \\ \frac{1}{|p|} a^{(1-p)/p} & \text{if } p < 0. \end{cases}$$

*Proof.* This is a standard result in matrix analysis (e.g., (Bhatia, 2013)). □

**Lemma C.2** (Weyl's inequality and Dual Weyl's inequality). *Let $A, B \in \mathbb{S}_n$, and $\lambda_1 \geq \lambda_2 \geq \cdots \geq \lambda_n$ denote the ordered eigenvalues. Weyl's inequality and the dual Weyl's inequality state that:*

$$\lambda_{i+j-1}(A+B) \leq \lambda_i(A) + \lambda_j(B), \ i,j \geq 1, \ i+j-1 \leq n, \quad \lambda_{i+j-n}(A+B) \geq \lambda_i(A) + \lambda_j(B), \ i,j \geq 1, \ i+j-n \geq 1.$$

*Hence, we obtain:*

$$|\lambda_i(A) - \lambda_i(B)| \leq \|A - B\|_2, \qquad 1 \leq i \leq n.$$

*Proof.* Let $E := A - B$, so $A = B + E$; applying Weyl's inequality with $j = 1$ and the dual Weyl inequality with $j = n$ gives $\lambda_i(B) + \lambda_i(E) \leq \lambda_i(A) \leq \lambda_i(B) + \lambda_1(E)$. Hence, for all $i$, we get:

$$|\lambda_i(A) - \lambda_i(B)| \leq \max\{|\lambda_1(E)|, |\lambda_n(E)|\} = \|E\|_2 = \|A - B\|_2.$$

□

**Lemma C.3** (Variant of Davis-Kahan Theorem). *Let $A, B \in \mathbb{S}_m$, and eigenvalues $\hat{\lambda}_1 \geq \cdots \geq \hat{\lambda}_m$ and $\lambda_1 \geq \cdots \geq \lambda_m$ respectively. Fix $1 \leq r \leq s \leq m$ and assume that $\min(\lambda_{r-1} - \lambda_r, \lambda_s - \lambda_{s+1}) > 0$, where $\lambda_0 := \infty$, $\lambda_{m+1} := -\infty$. Let $d := s - r + 1$ and let $V = [v_r, \cdots, v_s] \in \mathbb{R}^{m \times d}$, $\hat{V} = [\hat{v}_r, \cdots, \hat{v}_s] \in \mathbb{R}^{m \times d}$ have orthonormal columns satisfying $Av_j = \lambda_j v_j$ and $B\hat{v}_j = \hat{\lambda}_j \hat{v}_j$, for $r \leq j \leq s$. Then:*

$$\|\sin\Theta(V, \hat{V})\|_F \leq \frac{2\min(d^{1/2}\|B - A\|_2, \|B - A\|_F)}{\min(\lambda_{r-1} - \lambda_r, \lambda_s - \lambda_{s+1})},$$

*where $\sin\Theta$ is the diagonal matrix of the sines of the principal angles.*

*Proof.* See Theorem 2 of Yu et al. (2015). □

**Lemma C.4** (FTL-BTL (Kalai & Vempala, 2005) with discount weights). *Fix $\beta \in (0, 1)$ and horizon $T \in \mathbb{N}$. Let $\Phi : \mathcal{S}_{d,+} \to \mathbb{R}$ be any function (typically convex). For $t \geq 0$, define*

$$H_t \in \arg\min_{H \succ 0} \left\{ \Phi(H) + \sum_{s=1}^t \left\langle (1-\beta)\beta^{T-s} g_s g_s^\top, \ H^{-1} \right\rangle_F \right\}.$$

*(Equivalently, let $S_t^{(T)} := (1-\beta)\sum_{s=1}^t \beta^{T-s} g_s g_s^\top$ and $H_t \in \arg\min_{H \succ 0}\{\langle S_t^{(T)}, H^{-1}\rangle_F + \Phi(H)\}$.) Then*

$$(1-\beta)\sum_{t=1}^T \beta^{T-t} \|g_t\|_{H_t}^{\star 2} \leq (1-\beta)\sum_{t=1}^T \beta^{T-t} \|g_t\|_{H_T}^{\star 2} + \Phi(H_T) - \Phi(H_0).$$

**Lemma C.5.** *Let $G \in \mathbb{R}^{m \times n}$ be a matrix of rank at most $r$ and denote $g = \mathrm{vec}(G)$. Then:*

$$\frac{1}{r} g g^T \preceq \mathbf{I}_m \otimes (G^T G) \quad and \quad \frac{1}{r} g g^T \preceq (G G^T) \otimes \mathbf{I}_n.$$

*Proof.* This is Lemma 9 of Gupta et al. (2017). □

**Lemma C.6** (Monotonicity of the matrix geometric mean). *Let $A, B, C, D \in \mathbb{S}_+^d$ with*

$$A \preceq C, \qquad B \preceq D.$$

*Then*

$$A \# B \preceq C \# D,$$

*where*

$$A \# B := A^{1/2} \left( A^{-1/2} B A^{-1/2} \right)^{1/2} A^{1/2}.$$

*In particular, if $X \preceq P$ and $X \preceq Q$, then*

$$X = X \# X \preceq P \# Q.$$

*If moreover $P$ and $Q$ commute, then*

$$P \# Q = P^{1/2} Q^{1/2}.$$

**Lemma C.7.** *Assume that $G_1, \cdots, G_T \in \mathbb{R}^{m \times n}$ are matrices of rank at most $r$. Let $g_t = \mathrm{vec}(G_t)$ for all $1 \le t \le T$. Then, for all $\epsilon > 0$ and $0 < \beta < 1$:*

$$\epsilon \mathbf{I}_{mn} + \frac{1-\beta}{r} \sum_{t=1}^{T} \beta^{T-t} g_t g_t^T \preceq \left( \epsilon \mathbf{I}_m + (1-\beta) \sum_{t=1}^{T} \beta^{T-t} G_t G_t^T \right)^{1/2} \otimes \left( \epsilon \mathbf{I}_n + (1-\beta) \sum_{t=1}^{T} \beta^{T-t} G_t^T G_t \right)^{1/2}.$$

*Proof.* Define

$$A_n := \epsilon \mathbf{I}_n + (1-\beta) \sum_{t=1}^{T} \beta^{T-t} G_t^\top G_t, \qquad B_m := \epsilon \mathbf{I}_m + (1-\beta) \sum_{t=1}^{T} \beta^{T-t} G_t G_t^\top,$$

and

$$X := \epsilon \mathbf{I}_{mn} + \frac{1-\beta}{r} \sum_{t=1}^{T} \beta^{T-t} g_t g_t^\top.$$

From Lemma C.5, for each $t$,

$$\frac{1}{r} g_t g_t^\top \preceq \mathbf{I}_m \otimes (G_t^\top G_t), \qquad \frac{1}{r} g_t g_t^\top \preceq (G_t G_t^\top) \otimes \mathbf{I}_n.$$

Summing with weights $(1-\beta)\beta^{T-t}$ and adding $\epsilon \mathbf{I}_{mn}$ gives

$$X \preceq \mathbf{I}_m \otimes A_n =: P, \qquad X \preceq B_m \otimes \mathbf{I}_n =: Q.$$

Since $\epsilon > 0$, we have $A_n \succ 0$ and $B_m \succ 0$, hence $P, Q \in \mathbb{S}_+^{mn}$. Therefore, by Lemma C.6,

$$X = X \# X \preceq P \# Q.$$

Moreover,

$$P Q = (\mathbf{I}_m \otimes A_n)(B_m \otimes \mathbf{I}_n) = B_m \otimes A_n = (B_m \otimes \mathbf{I}_n)(\mathbf{I}_m \otimes A_n) = Q P,$$

so $P$ and $Q$ commute. Hence

$$P \# Q = P^{1/2} Q^{1/2}.$$

Using Lemma C.8,

$$P^{1/2}Q^{1/2} = (\mathbf{I}_m \otimes A_n)^{1/2}(B_m \otimes \mathbf{I}_n)^{1/2} = (\mathbf{I}_m \otimes A_n^{1/2})(B_m^{1/2} \otimes \mathbf{I}_n) = B_m^{1/2} \otimes A_n^{1/2}.$$

Therefore,

$$X \preceq B_m^{1/2} \otimes A_n^{1/2}.$$

□

**Lemma C.8** (Kronecker product properties). *We summarize well-known properties of the Kronecker product. Let $A, B, C, D$ be appropriately dimensioned matrices.*

1. $\|A \otimes B\| = \|A\|\|B\|$, where $\|\cdot\|$ is the spectral or Frobenius norm;

2. $(A \otimes B)^k = A^k \otimes B^k$, for all $k \in \mathbb{R}$, where $A, B \in \mathbb{S}_{n,+}$;

3. $(A + B) \otimes C = (A \otimes C) + (B \otimes C)$;

4. $k(A \otimes B) = A \otimes (kB) = (kA) \otimes B$, for all $k \in \mathbb{R}$;

5. $(A \otimes B)(C \otimes D) = (AC \otimes BD)$.

6. $\langle A, B \rangle = \text{vec}(A)^\top \text{vec}(B)$.

**Lemma C.9** (Operator monotone property). *If $0 \preceq A \preceq B$, then $A^{1/k} \preceq B^{1/k}$ for $k \geq 1$.*

**Lemma C.10.** *Given $A \in \mathbb{S}_{m,+}$:*

$$\det(A) \leq \left(\frac{\text{Tr}(A)}{m}\right)^m.$$

*Proof.* $\det(A) = \prod_{i=1}^m \lambda_i(A)$ and $\text{Tr}(A) = \sum_{i=1}^m \lambda_i(A)$; apply the AM-GM inequality. □

**Lemma C.11** (Daleckii–Kreĭn Theorem (Noferini, 2016)). *Let $X \in \mathbb{S}^n$ have the eigendecomposition $X = U\Lambda U^\top$ with $\Lambda = \text{diag}(\lambda_1, \ldots, \lambda_n)$, and let $f : \mathbb{R} \to \mathbb{R}$ be continuously differentiable on an open interval containing the spectrum $\{\lambda_i\}_{i=1}^n$. Define the spectral matrix function $f(X) := Uf(\Lambda)U^\top$, where $f(\Lambda) = \text{diag}(f(\lambda_1), \ldots, f(\lambda_n))$. Then, the Fréchet derivative of $f$ at $X$ in the direction $E \in \mathbb{S}^n$ satisfies*

$$Df(X)[E] = U\left(f^{[1]}(\Lambda) \circ (U^\top EU)\right)U^\top, \tag{4}$$

*where $\circ$ denotes the Hadamard product and the matrix of divided differences $f^{[1]}$ is defined by*

$$f_{ij}^{[1]} = \begin{cases} \dfrac{f(\lambda_i) - f(\lambda_j)}{\lambda_i - \lambda_j}, & i \neq j, \\ f'(\lambda_i), & i = j. \end{cases}$$

## D. Derivation of Staleness-Induced Preconditioner Gap

In this section, we provide a derivation process for the following equation:

$$P_t - \hat{P}_t = \{(L_t + \epsilon\mathbf{I}_m)^{-1/p} \otimes (R_t + \epsilon\mathbf{I}_n)^{-1/p}\} - (\hat{L}_t^{-1/p} \otimes \hat{R}_t^{-1/p})$$
$$= \hat{L}_t^{-1/p} \otimes \Delta_R(t) + \Delta_L(t) \otimes \hat{R}_t^{-1/p} + \Delta_L(t) \otimes \Delta_R(t)$$

*Proof.* Note that $(L_t + \epsilon\mathbf{I}_m)^{-1/p} = \Delta_L(t) + \hat{L}_t^{-1/p}$, and $(R_t + \epsilon\mathbf{I}_n)^{-1/p} = \Delta_R(t) + \hat{R}_t^{-1/p}$. By the property of the Kronecker product (Lemma C.8), we get:

$$(\Delta_L(t) + \hat{L}_t^{-1/p}) \otimes (\Delta_R(t) + \hat{R}_t^{-1/p}) = (\Delta_L(t) \otimes \Delta_R(t)) + (\Delta_L(t) \otimes \hat{R}_t^{-1/p}) + (\hat{L}_t^{-1/p} \otimes \Delta_R(t)) + (\hat{L}_t^{-1/p} \otimes \hat{R}_t^{-1/p}).$$

Hence, we obtain the desired result. □

## E. Proof of Operator-Gap Staleness Error Bound (Lemma 5.2)

**Lemma E.1.** *Under Assumption 5.1, for all $\epsilon > 0$, $p \geq 1$, and $1 \leq t \leq T$, with probability at least $1 - \delta$:*

$$\|\Delta_L(t)\|_2 \leq \frac{2}{p\epsilon^{(p+1)/p}}(1 - \beta^{\mathbf{f}})R_{SG}^2,$$

$$\|\Delta_R(t)\|_2 \leq \frac{2}{p\epsilon^{(p+1)/p}}(1 - \beta^{\mathbf{f}})R_{SG}^2,$$

*where*

$$R_{SG} := R + K\sigma\sqrt{\log(T/\delta)},$$

*with $0 < \delta < 1$ and a universal constant $K > 0$.*

*Proof.* **Step 1. (Good event)** Define

$$\mathcal{E} := \left\{ \max_{1 \leq t \leq T} \|G_t\|_2 \leq R_{SG} \right\}.$$

By the union bound,

$$\mathbb{P}(\mathcal{E}^c) \leq \sum_{t=1}^{T} \mathbb{P}(\|G_t\|_2 > R_{SG}).$$

By Assumption 5.1,

$$\|G_t\|_2 \leq \|\mathbb{E}[G_t \mid \mathcal{F}_{t-1}]\|_2 + \|G_t - \mathbb{E}[G_t \mid \mathcal{F}_{t-1}]\|_2$$
$$\leq R + \|G_t - \mathbb{E}[G_t \mid \mathcal{F}_{t-1}]\|_2,$$

so

$$\{\|G_t\|_2 > R_{SG}\} \subseteq \{\|G_t - \mathbb{E}[G_t \mid \mathcal{F}_{t-1}]\|_2 > R_{SG} - R\}.$$

Therefore, by the tail bound in Assumption 5.1,

$$\mathbb{P}(\|G_t\|_2 > R_{SG}) \leq \frac{\delta}{T},$$

hence $\mathbb{P}(\mathcal{E}) \geq 1 - \delta$.

**Step 2. (Bounding $\|L_t\|_2, \|R_t\|_2$ on $\mathcal{E}$)** Under $\mathcal{E}$, $\|G_t\|_2 \leq R_{SG}$ for all $t$, hence $\|G_t G_t^\top\|_2 \leq R_{SG}^2$. We show $\|L_t\|_2 \leq R_{SG}^2$ by induction (the same holds for $R_t$). Since $L_0 = \mathbf{0}_m \preceq R_{SG}^2 \mathbf{I}_m$, the base case holds. Assume $L_{t-1} \preceq R_{SG}^2 \mathbf{I}_m$. Then

$$L_t = \beta L_{t-1} + (1 - \beta)G_t G_t^T \preceq \beta R_{SG}^2 \mathbf{I}_m + (1 - \beta)R_{SG}^2 \mathbf{I}_m = R_{SG}^2 \mathbf{I}_m,$$

so $\|L_t\|_2 \leq R_{SG}^2$.

**Step 3. (Correct stale index and bounding $\|L_t - \widehat{L}_t\|_2$)** Recall that the algorithm refreshes the preconditioner only every $\mathbf{f}$ steps. Note that the stale Kronecker statistics $\hat{L}_t = L_{\tau(t)}, \hat{R}_t := R_{\tau(t)}$. For stale Shampoo, $\mathcal{C} = C_{\mathbf{f}}$, hence $\tau(t) = 1 + \mathbf{f}\lfloor(t-1)/f\rfloor$. Let $\ell = t - \tau(t) \in \{0, 1, \cdots, \mathbf{f} - 1\}$. If $\ell = 0$, then $L_t = \hat{L}_t$, and the claim is trivial; hence, assume $\ell \geq 1$. Unrolling the EMA recursion from $\tau(t)$ to $t$ yields

$$L_t = \beta^\ell L_{\tau(t)} + (1 - \beta) \sum_{k=\tau(t)+1}^{t} \beta^{t-k} G_k G_k^\top,$$

so

$$\|L_t - \widehat{L}_t\|_2 \leq (1 - \beta^\ell)\|L_{\tau(t)}\|_2 + (1 - \beta) \sum_{k=\tau(t)+1}^{t} \beta^{t-k} \|G_k G_k^\top\|_2$$

$$\leq (1 - \beta^\ell)R_{SG}^2 + (1 - \beta) \sum_{j=0}^{\ell-1} \beta^j R_{SG}^2 = 2(1 - \beta^\ell)R_{SG}^2$$

$$\leq 2(1 - \beta^{\mathbf{f}})R_{SG}^2.$$

Hence,

$$\|L_t - \widehat{L}_t\|_2 \le 2(1 - \beta^{\mathbf{f}})R_{\text{SG}}^2 \tag{5}$$

holds on $\mathcal{E}$. The same logic holds for $\|R_t - \widehat{R}_t\|_2$.

**Step 4. (Lipschitz bound and Frobenius norm)** Apply Lemma C.1 to the scalar function $f(\lambda) = \lambda^{-1/p}$ on $[\epsilon, \epsilon + R_{\text{SG}}^2]$. Since $|f'(\lambda)| = \frac{1}{p}\lambda^{-(p+1)/p}$, we have $\sup_{\lambda \in [\epsilon, \epsilon + R_{\text{SG}}^2]} |f'(\lambda)| = \frac{1}{p\epsilon^{(p+1)/p}}$. Thus, on $\mathcal{E}$:

$$\|\Delta_L(t)\|_2 = \|(L_t + \epsilon\mathbf{I}_m)^{-1/p} - (\widehat{L}_t + \epsilon\mathbf{I}_m)^{-1/p}\|_2$$
$$\le \frac{1}{p\epsilon^{(p+1)/p}} \|L_t - \widehat{L}_t\|_2 \le \frac{2}{p\epsilon^{(p+1)/p}}(1 - \beta^{\mathbf{f}})R_{\text{SG}}^2,$$

and similarly

$$\|\Delta_R(t)\|_2 \le \frac{2}{p\epsilon^{(p+1)/p}}(1 - \beta^{\mathbf{f}})R_{\text{SG}}^2.$$

We obtain the desired result. $\qquad\square$

## F. Proof of staleness-Induced Eigenvalue Gap and Eigenspace Rotation Bound

**Theorem F.1.** *Let the eigenvalues of $(L_t + \epsilon \mathbf{I}_m)^{-1/p}$ and $(R_t + \epsilon \mathbf{I}_n)^{-1/p}$ be sorted in non-increasing order, denoted $\lambda_1(\cdot) \geq \lambda_2(\cdot) \geq \cdots$ for all $p \geq 1$. Then, under Assumption 5.1, with probability at least $1 - \delta$, for all $0 \leq t \leq T$:*

$$|\lambda_i((L_t + \epsilon\mathbf{I}_m)^{-1/p}) - \lambda_i(\widehat{L}_t^{-1/p})| \leq \|(L_t + \epsilon\mathbf{I}_m)^{-1/p} - \widehat{L}_t^{-1/p}\|_2$$
$$\leq \frac{2(1 - \beta^{\mathbf{f}})R_{SG}^2}{p\epsilon^{(p+1)/p}},$$

$$|\lambda_i((R_t + \epsilon\mathbf{I}_n)^{-1/p}) - \lambda_i(\widehat{R}_t^{-1/p})| \leq \|(R_t + \epsilon\mathbf{I}_n)^{-1/p} - \widehat{R}_t^{-1/p}\|_2$$
$$\leq \frac{2(1 - \beta^{\mathbf{f}})R_{SG}^2}{p\epsilon^{(p+1)/p}}.$$

*Proof.* By Lemma C.2 and Lemma C.1 (applied to $f(\lambda) = \lambda^{-1/p}$), we obtain the claim. □

**Theorem F.2.** *Let $A := (L_{\tau(t)} + \epsilon\mathbf{I}_m)^{-1/p}$ and $\widehat{A} := (L_t + \epsilon\mathbf{I}_m)^{-1/p}$, with eigenvalues $\lambda_1 \geq \lambda_2 \geq \cdots \geq \lambda_{m-1} > \lambda_m$ and $\hat{\lambda}_1 \geq \hat{\lambda}_2 \geq \cdots \geq \hat{\lambda}_{m-1} > \hat{\lambda}_m$ respectively. Let $V_\ell := [v_1 \cdots v_{m-1}] \in \mathbb{R}^{m \times (m-1)}$, $\hat{V}_\ell := [\hat{v}_1 \cdots \hat{v}_{m-1}] \in \mathbb{R}^{m \times (m-1)}$ have orthonormal columns satisfying $Av_j = \lambda_j v_j$ and $\widehat{A}\hat{v}_j = \hat{\lambda}_j \hat{v}_j$ for $1 \leq j \leq m - 1$. Then, under Assumption 5.1, with probability at least $1 - \delta$, for all $0 \leq t \leq T$:*

$$\|\sin\Theta(V_\ell, \hat{V}_\ell)\|_F \leq \frac{2\min\left(\sqrt{m-1}\|\widehat{A} - A\|_2, \|\widehat{A} - A\|_F\right)}{\lambda_{m-1} - \lambda_m}$$
$$\leq \frac{4\sqrt{m}(1 - \beta^{\mathbf{f}})R_{SG}^2}{p\epsilon^{(p+1)/p}(\lambda_{m-1} - \lambda_m)}.$$

*The case of $A_R := (R_{\tau(t)} + \epsilon\mathbf{I}_n)^{-1/p}$ and $\widehat{A}_R := (R_t + \epsilon\mathbf{I}_n)^{-1/p}$ is analogous.*

*Proof.* Let $\star := \widehat{A} - A$. The first inequality follows from Lemma C.3. Next,

$$\min\left(\sqrt{m-1}\|\star\|_2, \|\star\|_F\right) \leq \sqrt{m}\|\star\|_2.$$

By Lemma C.1 (applied to $f(\lambda) = \lambda^{-1/p}$),
$$\|\star\|_2 \leq \frac{2(1 - \beta^{\mathbf{f}})R_{SG}^2}{p\epsilon^{(p+1)/p}}.$$

Substituting yields the result. □

## G. Proof of Discounted Regret Bound (Theorem 5.4)

**Theorem G.1.** *Let us assume that Assumption 5.1 and Assumption 5.3 hold. Furthermore, assume* $\text{rank}(G_t) \leq r \leq \min(m,n)$ *and* $\max_t \|W_t - W^\star\|_F = D(W^\star) =: D$ *for all* $W^\star \in \mathbb{R}^{m \times n}$. *Then, for all* $\epsilon > 0$, *with probability at least* $1 - \delta$, *the (discount) regret bound of stale Shampoo with* $p = 2$ *is:*

$$\sum_{t=1}^{T} \beta^{T-t} f_t(W_t) - \sum_{t=1}^{T} \beta^{T-t} f_t(W^\star) \leq \frac{1}{2\eta} \left\{ \epsilon D^2 + \frac{\sqrt{R_{SG}^2 + \epsilon} D^2}{\sqrt{\epsilon}} R_{SG}^2 + \frac{D^2(R_{SG}^2 + \epsilon)}{\beta} \right\}$$
$$+ \frac{\eta}{1-\beta} \left\{ \frac{r(1-\beta^{\mathbf{f}})R_{SG}^4}{\epsilon^2} \right\} + \frac{\eta}{2(1-\beta)} \left\{ rmn \log\left( \epsilon + \frac{(1-\beta^T)rR_{SG}^2}{mn} \right) - rmn \log(\epsilon) \right\},$$

*and stale Shampoo with* $p = 4$ *is:*

$$\sum_{t=1}^{T} \beta^{T-t} f_t(W_t) - \sum_{t=1}^{T} \beta^{T-t} f_t(W^\star) \leq \frac{1}{2\eta} \left\{ \sqrt{\epsilon}D^2 + \frac{\sqrt[4]{R_{SG}^2 + \epsilon} D^2}{2\epsilon^{3/4}} R_{SG}^2 + \frac{D^2 \sqrt{R_{SG}^2 + \epsilon}}{\beta} \right\}$$
$$+ \underbrace{\frac{\eta r(1-\beta^{\mathbf{f}})R_{SG}^4}{2(1-\beta)\epsilon^{3/2}}}_{\text{Regret staleness error}} + \frac{\eta mnr \sqrt{R_{SG}^2 + \epsilon}}{1-\beta}.$$

*Proof.* We proceed under the good event $\mathcal{E}$, which holds with probability at least $1 - \delta$.

**Preconditioners and phases.** For $t \geq 1$ and $p \geq 1$, define

$$H_{t,p} = (L_t + \epsilon \mathbf{I}_m)^{1/p} \otimes (R_t + \epsilon \mathbf{I}_n)^{1/p}, \qquad (L_0 = \mathbf{0}_m, \ R_0 = \mathbf{0}_n).$$

Assume for simplicity that $T = \mathbf{f}K$ (otherwise, take $K = \lceil T/\mathbf{f} \rceil$ and the last phase is shorter; the same bounds apply). Define phase indices $T_k = (k-1)\mathbf{f} + 1$ and phases $t \in \{T_k, \ldots, T_{k+1} - 1\}$. Define for each phase $k$:

$$L_k := L_{T_k}, \qquad R_k := R_{T_k}, \qquad H_{k,p} := (L_k + \epsilon \mathbf{I}_m)^{1/p} \otimes (R_k + \epsilon \mathbf{I}_n)^{1/p}.$$

For $t$ in phase $k$, stale Shampoo uses the fixed preconditioner $H_{k,p}$.

**Step 1.** Let $w_t = \text{vec}(W_t), g_t = \text{vec}(G_t)$. Multiplying by $(1 - \beta)$, applying convexity and Lemma C.8, we have:

$$(1-\beta)\sum_{t=1}^{T} \beta^{T-t} \big( f_t(W_t) - f_t(W^\star) \big) \leq (1-\beta)\sum_{t=1}^{T} \beta^{T-t} \langle G_t, W_t - W^\star \rangle = (1-\beta)\sum_{t=1}^{T} \beta^{T-t} g_t^\top (w_t - w^\star).$$

By Lemma 3 of Gupta et al. (2017),

$$(1-\beta)\sum_{t=1}^{T} \beta^{T-t} g_t^\top (w_t - w^\star) \leq \frac{1}{2\eta} \underbrace{(1-\beta)\sum_{t=1}^{T} \beta^{T-t} \Delta_t(w^\star)}_{(a)} + \frac{\eta}{2} \underbrace{(1-\beta)\sum_{t=1}^{T} \beta^{T-t} \|g_t\|_{H_{t,p}}^{\star 2}}_{(b)},$$

where $\Delta_t(x^\star) := \|x_t - x^\star\|_{H_{t,p}}^2 - \|x_{t+1} - x^\star\|_{H_{t,p}}^2$.

**Step 2: Bounding** $(a)$. Since $H_{t,p} = H_{k(t),p}$ within each phase,

$$(a) = (1-\beta)\sum_{k=1}^{K} \sum_{t=T_k}^{T_{k+1}-1} \beta^{T-t} \Big( \|w_t - w^\star\|_{H_{k,p}}^2 - \|w_{t+1} - x^\star\|_{H_{k,p}}^2 \Big).$$

Let $k(t)$ be the phase index containing $t$ and define $A_t := \|w_t - w^\star\|_{H_{k(t),p}}^2$ and $\widetilde{A}_{t+1} := \|w_{t+1} - w^\star\|_{H_{k(t),p}}^2$. Then

$$(a) = (1-\beta)\sum_{t=1}^{T} \beta^{T-t}(A_t - \widetilde{A}_{t+1}).$$

Add and subtract $A_{t+1}$:

$$A_t - \widetilde{A}_{t+1} = (A_t - A_{t+1}) + (A_{t+1} - \widetilde{A}_{t+1}),$$

hence

$$(a) = (1-\beta)\sum_{t=1}^{T}\beta^{T-t}(A_t - A_{t+1}) + (1-\beta)\sum_{t=1}^{T}\beta^{T-t}(A_{t+1} - \widetilde{A}_{t+1}).$$

**(i) Discounted telescoping term.** For any sequence $\{A_t\}$,

$$(1-\beta)\sum_{t=1}^{T}\beta^{T-t}(A_t - A_{t+1}) = (1-\beta)\big[\beta^{T-1}A_1 - \beta^{-1}A_{T+1}\big] + (1-\beta)^2\sum_{t=2}^{T}\beta^{T-t}A_t.$$

Dropping the negative term $-\beta^{-1}A_{T+1}$ and using $\|w_t - w^\star\|_2 \le D$ and by Lemma C.8, we get $\|H_{k(t),p}\|_2 \le (R_{\text{SG}}^2 + \epsilon)^{2/p}$ on $\mathcal{E}$, hence:

$$
\begin{aligned}
(1-\beta)\sum_{t=1}^{T}\beta^{T-t}(A_t - A_{t+1}) &\le (1-\beta)\beta^{T-1}\|w_1 - x^\star\|_{H_{1,p}}^2 + (1-\beta)^2\sum_{t=2}^{T}\beta^{T-t}\|w_t - w^\star\|_{H_{k(t),p}}^2 \\
&\le (1-\beta)\epsilon^{2/p}D^2 + (1-\beta)^2 D^2 (R_{\text{SG}}^2 + \epsilon)^{2/p}\sum_{t=2}^{T}\beta^{T-t} \\
&\le (1-\beta)\epsilon^{2/p}D^2 + (1-\beta)\frac{1-\beta^T}{\beta}D^2(R_{\text{SG}}^2 + \epsilon)^{2/p}.
\end{aligned}
$$

**(ii) Phase-boundary correction term.** Note $A_{t+1} - \widetilde{A}_{t+1} = 0$ unless $t = T_k - 1$ for some $k \ge 2$, in which case

$$A_{T_k} - \widetilde{A}_{T_k} = \|w_{T_k} - w^\star\|_{H_{k,p}}^2 - \|w_{T_k} - w^\star\|_{H_{k-1,p}}^2.$$

Therefore,

$$
\begin{aligned}
(1-\beta)\sum_{t=1}^{T}\beta^{T-t}(A_{t+1} - \widetilde{A}_{t+1}) &= (1-\beta)\sum_{k=2}^{K}\beta^{T-(T_k-1)}\Big(\|w_{T_k} - w^\star\|_{H_{k,p}}^2 - \|w_{T_k} - w^\star\|_{H_{k-1,p}}^2\Big) \\
&\le (1-\beta)\sum_{k=2}^{K}\beta^{T-T_k}D^2\,\|H_{k,p} - H_{k-1,p}\|_2,
\end{aligned}
$$

where we used $\beta^{T-(T_k-1)} = \beta\beta^{T-T_k} \le \beta^{T-T_k}$ since $\beta \in (0,1)$. On $\mathcal{E}$, for $k \ge 2$, we have:

$$\|L_k - L_{k-1}\|_2 \le 2(1-\beta^{\mathbf{f}})R_{\text{SG}}^2, \qquad \|R_k - R_{k-1}\|_2 \le 2(1-\beta^{\mathbf{f}})R_{\text{SG}}^2.$$

via Lemma 5.2.

Using Lemma C.1, Lemma C.8 and the inequality

$$\|A \otimes B - C \otimes D\|_2 \le \|A - C\|_2\|B\|_2 + \|C\|_2\|B - D\|_2,$$

we obtain

$$
\begin{aligned}
\|H_{k,p} - H_{k-1,p}\|_2 &= \|(L_k + \epsilon\mathbf{I}_m)^{1/p} \otimes (R_k + \epsilon\mathbf{I}_n)^{1/p} - (L_{k-1} + \epsilon\mathbf{I}_m)^{1/p} \otimes (R_{k-1} + \epsilon\mathbf{I}_n)^{1/p}\|_2 \\
&\le \|(L_k + \epsilon\mathbf{I}_m)^{1/p} - (L_{k-1} + \epsilon\mathbf{I}_m)^{1/p}\|_2\|(R_k + \epsilon\mathbf{I}_n)^{1/p}\|_2 \\
&\quad + \|(L_{k-1} + \epsilon\mathbf{I}_m)^{1/p}\|_2\|(R_k + \epsilon\mathbf{I}_n)^{1/p} - (R_{k-1} + \epsilon\mathbf{I}_n)^{1/p}\|_2 \\
&\le \frac{1}{p}\epsilon^{(1-p)/p}\|L_k - L_{k-1}\|_2\,(R_{\text{SG}}^2 + \epsilon)^{1/p} + \frac{1}{p}\epsilon^{(1-p)/p}\|R_k - R_{k-1}\|_2\,(R_{\text{SG}}^2 + \epsilon)^{1/p} \\
&\le \frac{4(1-\beta^{\mathbf{f}})}{p}\,\epsilon^{(1-p)/p}\,(R_{\text{SG}}^2 + \epsilon)^{1/p}\,R_{\text{SG}}^2.
\end{aligned}
$$

Hence

$$(1-\beta)\sum_{t=1}^{T}\beta^{T-t}(A_{t+1}-\widetilde{A}_{t+1}) \le (1-\beta)D^2\frac{4(1-\beta^{\mathbf{f}})}{p}\epsilon^{(1-p)/p}(R_{\mathrm{SG}}^2+\epsilon)^{1/p}R_{\mathrm{SG}}^2\sum_{k=2}^{K}\beta^{T-T_k}.$$

Finally,

$$\sum_{k=2}^{K}\beta^{T-T_k} = \sum_{k=2}^{K}\beta^{K\mathbf{f}-((k-1)\mathbf{f}+1)} = \beta^{-1}\sum_{k=2}^{K}(\beta^{\mathbf{f}})^{K-k+1} \le \frac{\beta^{\mathbf{f}-1}}{1-\beta^{\mathbf{f}}}.$$

Therefore,

$$(1-\beta)\sum_{t=1}^{T}\beta^{T-t}(A_{t+1}-\widetilde{A}_{t+1}) \le \beta^{\mathbf{f}-1}\frac{4(1-\beta)}{p}\,\epsilon^{(1-p)/p}(R_{\mathrm{SG}}^2+\epsilon)^{1/p}D^2R_{\mathrm{SG}}^2.$$

**Conclusion for** $(a)$**.** Combining (i) and (ii):

$$(a) \le (1-\beta)\epsilon^{2/p}D^2 + \beta^{\mathbf{f}-1}\frac{4(1-\beta)}{p}\,\epsilon^{(1-p)/p}(R_{\mathrm{SG}}^2+\epsilon)^{1/p}D^2R_{\mathrm{SG}}^2 + (1-\beta)\frac{1-\beta^T}{\beta}\,D^2(R_{\mathrm{SG}}^2+\epsilon)^{2/p}.$$

**Step 3: Bounding** $(b)$**.** Write

$$(b) = (1-\beta)\Big\{\sum_{k=1}^{K}\sum_{t=T_k}^{T_{k+1}-1}\beta^{T-t}\Big(\|g_t\|_{H_{k,p}}^{\star 2} - \|g_t\|_{H_{t,p}}^{\star 2}\Big) + \sum_{t=1}^{T}\beta^{T-t}\|g_t\|_{H_{t,p}}^{\star 2}\Big\}. \tag{6}$$

We consider $p = 2$ and $p = 4$.

1. **Stale Shampoo with** $p = 2$**.**

   **(b1) Stale–fresh gap term.** For $t$ in phase $k$,

   $$\|g_t\|_{H_{k,2}}^{\star 2} - \|g_t\|_{H_{t,2}}^{\star 2} = g_t^\top\Big((H_{k,2})^{-1} - (H_{t,2})^{-1}\Big)g_t$$
   $$\le \|g_t\|_2^2\,\|(H_{k,2})^{-1} - (H_{t,2})^{-1}\|_2 \le rR_{\mathrm{SG}}^2\,\|(H_{k,2})^{-1} - (H_{t,2})^{-1}\|_2,$$

   where $\|g_t\|_2^2 = \|G_t\|_F^2 \le r\|G_t\|_2^2 \le rR_{\mathrm{SG}}^2$ on $\mathcal{E}$.

   Let $\widehat{L}_t = L_k$ and $\widehat{R}_t = R_k$ for $t$ in phase $k$. Then by Lemma 5.2 with $p = 2$:

   $$\|(L_t + \epsilon\mathbf{I}_m)^{-1/2} - (L_k + \epsilon\mathbf{I}_m)^{-1/2}\|_2 \le \frac{2(1-\beta^{\mathbf{f}})R_{\mathrm{SG}}^2}{2\,\epsilon^{3/2}} = \frac{(1-\beta^{\mathbf{f}})R_{\mathrm{SG}}^2}{\epsilon^{3/2}},$$

   and similarly for $R$. Using the Kronecker difference inequality and $\|(R_k + \epsilon\mathbf{I}_n)^{-1/2}\|_2 \le \epsilon^{-1/2}$, we obtain

   $$\|(H_{k,2})^{-1} - (H_{t,2})^{-1}\|_2 \le \frac{2(1-\beta^{\mathbf{f}})R_{\mathrm{SG}}^2}{\epsilon^2}.$$

   Therefore,

   $$(1-\beta)\sum_{k=1}^{K}\sum_{t=T_k}^{T_{k+1}-1}\beta^{T-t}\Big(\|g_t\|_{H_{k,2}}^{\star 2} - \|g_t\|_{H_{t,2}}^{\star 2}\Big) \le (1-\beta)\sum_{t=1}^{T}\beta^{T-t}\,rR_{\mathrm{SG}}^2\cdot\frac{2(1-\beta^{\mathbf{f}})R_{\mathrm{SG}}^2}{\epsilon^2}$$
   $$= \frac{2r(1-\beta^{\mathbf{f}})(1-\beta^T)R_{\mathrm{SG}}^4}{\epsilon^2}.$$

   **(b2) Fresh term via FTL–BTL + logdet.** We apply Lemma C.4 with

   $$\Phi(H) = \log\det(H) + \epsilon\,\mathrm{Tr}(H^{-1}), \qquad S_t := (1-\beta)\sum_{s=1}^{t}\beta^{T-s}g_s g_s^\top.$$

Define
$$\hat{H}_t \in \arg\min_{H \succ 0} \left\{ \langle S_t, H^{-1} \rangle_F + \log\det(H) + \epsilon\,\mathrm{Tr}(H^{-1}) \right\}.$$

Using standard matrix calculus,
$$\hat{H}_t = \epsilon\mathbf{I}_{mn} + S_t.$$

By Lemma C.4:
$$(1-\beta)\sum_{t=1}^{T}\beta^{T-t}\|g_t\|_{\hat{H}_t}^{\star 2} \leq (1-\beta)\sum_{t=1}^{T}\beta^{T-t}\|g_t\|_{\hat{H}_T}^{\star 2} + \Phi(\hat{H}_T) - \Phi(\hat{H}_0).$$

Moreover,
$$(1-\beta)\sum_{t=1}^{T}\beta^{T-t}\|g_t\|_{\hat{H}_T}^{\star 2} + \epsilon\,\mathrm{Tr}(\hat{H}_T^{-1}) = \langle S_T, \hat{H}_T^{-1} \rangle_F + \epsilon\,\mathrm{Tr}(\hat{H}_T^{-1}) = \langle \epsilon\mathbf{I}_{mn} + S_T, \hat{H}_T^{-1} \rangle_F = \mathrm{Tr}(\hat{H}_T\hat{H}_T^{-1}) = mn.$$

Since $\hat{H}_0 = \epsilon\mathbf{I}_{mn}$, we have $\log\det(\hat{H}_0) = mn\log\epsilon$ and $\epsilon\,\mathrm{Tr}(\hat{H}_0^{-1}) = mn$. Therefore,

$$(1-\beta)\sum_{t=1}^{T}\beta^{T-t}\|g_t\|_{\hat{H}_t}^{\star 2} \leq \log\det(\hat{H}_T) - mn\log\epsilon.$$

**Relating $\hat{H}_t$ and $H_{t,2}$.** For $t \leq T$, by Lemma C.7 applied to $\{G_s\}_{s \leq t}$,

$$\epsilon\mathbf{I}_{mn} + \frac{1}{r}S_t \preceq A_t^{1/2} \otimes B_t^{1/2}, \quad A_t := \epsilon\mathbf{I}_m + (1-\beta)\sum_{s=1}^{t}\beta^{T-s}G_sG_s^{\top}, \ B_t := \epsilon\mathbf{I}_n + (1-\beta)\sum_{s=1}^{t}\beta^{T-s}G_s^{\top}G_s.$$

Since $r \geq 1$,
$$\hat{H}_t = \epsilon\mathbf{I}_{mn} + S_t \preceq r\left(\epsilon\mathbf{I}_{mn} + \frac{1}{r}S_t\right) \preceq r(A_t^{1/2} \otimes B_t^{1/2}).$$

Also, using $\beta^{T-s} = \beta^{T-t}\beta^{t-s}$ and $\beta^{T-t} \leq 1$:
$$A_t = \epsilon\mathbf{I}_m + \beta^{T-t}L_t \preceq \epsilon\mathbf{I}_m + L_t, \qquad B_t = \epsilon\mathbf{I}_n + \beta^{T-t}R_t \preceq \epsilon\mathbf{I}_n + R_t.$$

By Lemma C.9 (square-root monotonicity) and Lemma C.8,
$$A_t^{1/2} \otimes B_t^{1/2} \preceq (\epsilon\mathbf{I}_m + L_t)^{1/2} \otimes (\epsilon\mathbf{I}_n + R_t)^{1/2} =: H_{t,2}.$$

Thus $\hat{H}_t \preceq rH_{t,2}$, hence $H_{t,2}^{-1} \preceq r\hat{H}_t^{-1}$ and
$$\|g_t\|_{H_{t,2}}^{\star 2} \leq r\|g_t\|_{\hat{H}_t}^{\star 2}.$$

Therefore,

$$(1-\beta)\sum_{t=1}^{T}\beta^{T-t}\|g_t\|_{H_{t,2}}^{\star 2} \leq r(1-\beta)\sum_{t=1}^{T}\beta^{T-t}\|g_t\|_{\hat{H}_t}^{\star 2} \leq r\log\det(\hat{H}_T) - rmn\log\epsilon.$$

**Bounding $\log\det(\hat{H}_T)$ via trace.** Since $\hat{H}_T = \epsilon\mathbf{I}_{mn} + S_T$,

$$\mathrm{Tr}(\hat{H}_T) = mn\epsilon + (1-\beta)\sum_{s=1}^{T}\beta^{T-s}\,\mathrm{Tr}(g_sg_s^{\top}) = mn\epsilon + (1-\beta)\sum_{s=1}^{T}\beta^{T-s}\|g_s\|_2^2$$
$$\leq mn\epsilon + (1-\beta)\sum_{s=1}^{T}\beta^{T-s}rR_{\mathrm{SG}}^2 = mn\epsilon + (1-\beta^T)rR_{\mathrm{SG}}^2,$$

where we used $\|g_s\|_2^2 = \|G_s\|_F^2 \le r\|G_s\|_2^2 \le rR_{\mathrm{SG}}^2$ on $\mathcal{E}$. By Lemma C.10,

$$\log \det(\hat{H}_T) \le mn \log \left( \frac{\mathrm{Tr}(\hat{H}_T)}{mn} \right) \le mn \log \left( \epsilon + \frac{(1 - \beta^T)rR_{\mathrm{SG}}^2}{mn} \right).$$

Hence

$$(1 - \beta) \sum_{t=1}^{T} \beta^{T-t} \|g_t\|_{H_{t,2}}^{\star 2} \le rmn \log \left( \epsilon + \frac{(1 - \beta^T)rR_{\mathrm{SG}}^2}{mn} \right) - rmn \log \epsilon.$$

**Combine** $(a)$ **and** $(b)$ **for** $p = 2$. Plug the bounds for $(a)$ and $(b)$ into $\frac{1}{2\eta}(a) + \frac{\eta}{2}(b)$ and use $1 - \beta^T \le 1$ to simplify; finally divide by $(1 - \beta)$ to obtain the stated regret bound.

2. **Stale Shampoo with** $p = 4$.

   **(b1) Stale–fresh gap term.** For $t$ in phase $k$:

   $$\|g_t\|_{H_{k,4}}^{\star 2} - \|g_t\|_{H_{t,4}}^{\star 2} \le \|g_t\|_2^2 \|(H_{k,4})^{-1} - (H_{t,4})^{-1}\|_2 \le rR_{\mathrm{SG}}^2 \|(H_{k,4})^{-1} - (H_{t,4})^{-1}\|_2.$$

   From Lemma 5.2 with $p = 4$,

   $$\|(L_t + \epsilon I_m)^{-1/4} - (L_k + \epsilon I_m)^{-1/4}\|_2 \le \frac{2(1 - \beta^{\mathbf{f}})R_{\mathrm{SG}}^2}{4\,\epsilon^{5/4}} = \frac{(1 - \beta^{\mathbf{f}})R_{\mathrm{SG}}^2}{2\epsilon^{5/4}},$$

   and similarly for $R$. Using $\|(R_k + \epsilon I_n)^{-1/4}\|_2 \le \epsilon^{-1/4}$ and the Kronecker difference inequality,

   $$\|(H_{k,4})^{-1} - (H_{t,4})^{-1}\|_2 \le \frac{(1 - \beta^{\mathbf{f}})R_{\mathrm{SG}}^2}{\epsilon^{3/2}}.$$

   Therefore,

   $$(1 - \beta) \sum_{k=1}^{K} \sum_{t=T_k}^{T_{k+1}-1} \beta^{T-t} \left( \|g_t\|_{H_{k,4}}^{\star 2} - \|g_t\|_{H_{t,4}}^{\star 2} \right) \le \frac{r(1 - \beta^{\mathbf{f}})(1 - \beta^T)R_{\mathrm{SG}}^4}{\epsilon^{3/2}}.$$

   **(b2) Fresh term via FTL–BTL (trace potential).** Apply Lemma C.4 with

   $$\Phi(H) = \mathrm{Tr}(H) + \epsilon r\,\mathrm{Tr}(H^{-1}), \qquad S_t = (1 - \beta) \sum_{s=1}^{t} \beta^{T-s} g_s g_s^{\top},$$

   and define

   $$\hat{H}_t \in \arg\min_{H \succ 0} \left\{ \langle S_t, H^{-1} \rangle_F + \mathrm{Tr}(H) + \epsilon r\,\mathrm{Tr}(H^{-1}) \right\}.$$

   First-order optimality yields

   $$\hat{H}_t = (\epsilon r \mathbf{I}_{mn} + S_t)^{1/2}.$$

   Lemma C.4 gives

   $$(1 - \beta) \sum_{t=1}^{T} \beta^{T-t} \|g_t\|_{\hat{H}_t}^{\star 2} \le (1 - \beta) \sum_{t=1}^{T} \beta^{T-t} \|g_t\|_{\hat{H}_T}^{\star 2} + \Phi(\hat{H}_T) - \Phi(\hat{H}_0).$$

   Using $\hat{H}_0 = (\epsilon r)^{1/2} \mathbf{I}_{mn}$ and $\hat{H}_T^2 = \epsilon r \mathbf{I}_{mn} + S_T$,

   $$(1 - \beta) \sum_{t=1}^{T} \beta^{T-t} \|g_t\|_{\hat{H}_T}^{\star 2} + \epsilon r\,\mathrm{Tr}(\hat{H}_T^{-1}) = \langle \epsilon r \mathbf{I}_{mn} + S_T, \hat{H}_T^{-1} \rangle_F = \mathrm{Tr}(\hat{H}_T^2 \hat{H}_T^{-1}) = \mathrm{Tr}(\hat{H}_T).$$

   Hence

   $$(1 - \beta) \sum_{t=1}^{T} \beta^{T-t} \|g_t\|_{\hat{H}_t}^{\star 2} \le 2\,\mathrm{Tr}(\hat{H}_T).$$

**Relating $\hat{H}_t$ and $H_{t,4}$.** By Lemma C.7 applied to $\{G_s\}_{s \le t}$,

$$\epsilon \mathbf{I}_{mn} + \frac{1}{r} S_t \preceq A_t^{1/2} \otimes B_t^{1/2} \quad \Rightarrow \quad \epsilon r \mathbf{I}_{mn} + S_t \preceq r(A_t^{1/2} \otimes B_t^{1/2}),$$

with the same $A_t, B_t$ as above. Taking square roots and using Lemma C.9 and Lemma C.8,

$$\hat{H}_t = (\epsilon r \mathbf{I}_{mn} + S_t)^{1/2} \preceq \sqrt{r} \, (A_t^{1/2} \otimes B_t^{1/2})^{1/2} = \sqrt{r} \, (A_t^{1/4} \otimes B_t^{1/4}).$$

Since $A_t \preceq \epsilon \mathbf{I}_m + L_t$ and $B_t \preceq \epsilon \mathbf{I}_n + R_t$,

$$A_t^{1/4} \otimes B_t^{1/4} \preceq (\epsilon \mathbf{I}_m + L_t)^{1/4} \otimes (\epsilon \mathbf{I}_n + R_t)^{1/4} =: H_{t,4}.$$

Therefore $\hat{H}_t \preceq \sqrt{r} \, H_{t,4}$, so $H_{t,4}^{-1} \preceq \sqrt{r} \, \hat{H}_t^{-1}$ and

$$\|g_t\|_{H_{t,4}}^{\star 2} \le \sqrt{r} \, \|g_t\|_{\hat{H}_t}^{\star 2}.$$

Thus

$$(1 - \beta) \sum_{t=1}^{T} \beta^{T-t} \|g_t\|_{H_{t,4}}^{\star 2} \le \sqrt{r} \, (1 - \beta) \sum_{t=1}^{T} \beta^{T-t} \|g_t\|_{\hat{H}_t}^{\star 2} \le 2\sqrt{r} \, \mathrm{Tr}(\hat{H}_T) \le 2r \, \mathrm{Tr}(H_{T,4}),$$

where the last inequality uses $\hat{H}_T \preceq \sqrt{r} \, H_{T,4}$ and monotonicity of trace on PSD.

Finally,

$$\mathrm{Tr}(H_{T,4}) = \mathrm{Tr}((L_T + \epsilon \mathbf{I}_m)^{1/4}) \, \mathrm{Tr}((R_T + \epsilon \mathbf{I}_n)^{1/4}) \le m(R_{\mathrm{SG}}^2 + \epsilon)^{1/4} \cdot n(R_{\mathrm{SG}}^2 + \epsilon)^{1/4} = mn\sqrt{R_{\mathrm{SG}}^2 + \epsilon},$$

so

$$(1 - \beta) \sum_{t=1}^{T} \beta^{T-t} \|g_t\|_{H_{t,4}}^{\star 2} \le 2mnr\sqrt{R_{\mathrm{SG}}^2 + \epsilon}.$$

**Combine** $(a)$ **and** $(b)$ **for** $p = 4$**.** Plug the bounds for $(a)$ and $(b)$ into $\frac{1}{2\eta}(a) + \frac{\eta}{2}(b)$, use $1 - \beta^T \le 1$, and finally divide by $(1 - \beta)$ to obtain the stated bound.

$\square$

## H. Proof of Approximation of Relative Operator Error

**Theorem H.1** (Relative operator error for inverse $p$-th Root). *Let $A \in \mathbb{S}_+^n$ be a positive definite matrix and $E \in \mathbb{S}^n$ be a perturbation matrix. For any $p > 0$, the relative error of the inverse $p$-th root is bounded by:*

$$\frac{\|(A+E)^{-1/p} - A^{-1/p}\|_F}{\|A^{-1/p}\|_F} \leq \frac{\|(A+E)^{-1/p} - A^{-1/p}\|_F}{\|A^{-1/p}\|_2} \approx \frac{\|\Delta_1\|_F}{\|A^{-1/p}\|_2} \leq \frac{1}{p}\|A^{-1/2}EA^{-1/2}\|_F,$$

*where $\approx$ denotes the first-order approximation.*

*Proof.* We want to show how much $(A+E)^{-1/p}$ differs from $A^{-1/p}$ when matrix $A$ undergoes perturbation $E$ to become $A + E$. Let $A = U\Lambda U^T$ be the eigendecomposition of $A$, where $\Lambda = \text{diag}(\mu_1, \ldots, \mu_n)$ has eigenvalues $\mu_i > 0$. Given $F(A) = A^{-1/p}$ and $\Delta = F(A+E) - F(A)$, we have:

$$F(A+E) = F(A) + \Delta = F(A) + \underbrace{D_F(A)[E]}_{\Delta_1} + R_2,$$

where $\Delta_1$ is the first-order term and $R_2$ is the remainder term of the Fréchet derivative. We rotate the perturbation $E$ and $\Delta_1$ into the eigenbasis:

$$\hat{E} = U^T E U, \quad \hat{\Delta}_1 = U^T \Delta_1 U.$$

Since the Frobenius norm is orthogonally invariant, we have $\|\Delta_1\|_F = \|\hat{\Delta}_1\|_F$ and $\|E\|_F = \|\hat{E}\|_F$. The first-order term $\Delta_1$ corresponds to the Fréchet derivative of the function $f(t) = t^{-1/p}$ applied to $E$. In the eigenbasis, the entries of $\hat{\Delta}_1$ are given by the divided differences (also known as the Daleckii-Krein Theorem; Lemma C.11):

$$[\hat{\Delta}_1]_{ij} = \frac{\mu_i^{-1/p} - \mu_j^{-1/p}}{\mu_i - \mu_j}[\hat{E}]_{ij}.$$

(Note: If $\mu_i = \mu_j$, the fraction converges to the derivative $f'(\mu_i)$). Now, recall the definition of the whitened perturbation $\tilde{E} = A^{-1/2}EA^{-1/2}$. In the rotated basis, its entries $[\hat{\tilde{E}}]_{ij}$ satisfy:

$$[\hat{\tilde{E}}]_{ij} = [U^T A^{-1/2} E A^{-1/2} U]_{ij} = \frac{[\hat{E}]_{ij}}{\sqrt{\mu_i \mu_j}} \implies [\hat{E}]_{ij} = \sqrt{\mu_i \mu_j}[\hat{\tilde{E}}]_{ij}.$$

Substituting this relation into the expression for $[\hat{\Delta}_1]_{ij}$, we obtain:

$$[\hat{\Delta}_1]_{ij} = \underbrace{\left( \frac{\mu_i^{-1/p} - \mu_j^{-1/p}}{\mu_i - \mu_j} \sqrt{\mu_i \mu_j} \right)}_{C_{ij}} [\hat{\tilde{E}}]_{ij}.$$

We now bound the coefficient $C_{ij}$. Let $q = 1/p > 0$ denote. Without loss of generality, assume that $x = \mu_i \leq \mu_j = rx$, with $r \geq 1$.

$$|C_{ij}| = x^{-q}\frac{1 - r^{-q}}{r - 1}\sqrt{r}.$$

If $H(r) = \frac{(1-r^{-q})\sqrt{r}}{r-1} \leq q$, then $\max_{i,j}|C_{i,j}| \leq 1/p(\min_k \mu_k)^{-1/p} = 1/p\|A^{-1/p}\|_2$. Let us define

$$G(r) = q(r-1) - H(r)(r-1)$$
$$= q(r-1) - \sqrt{r} + r^{\frac{1}{2}-q}.$$

Then, by simple calculation, we have: $G(1) = 0, G'(1) = 0$ and

$$G'(r) = q - \frac{1}{2}r^{-1/2} + \left(\frac{1}{2} - q\right)r^{-\frac{1}{2}-q}$$
$$G''(r) = r^{-\frac{3}{2}-q}\left(\frac{1}{4}r^q - \frac{1}{4} + q^2\right).$$

Hence, we have $G''(r) > 0$ for all $r \geq 1$, and this gives a result $G(r) \geq 0$. Lastly, by L'Hôpital's rule, we have $\lim_{r \to 1} H(r) = q$. Therefore, we have $H(r) \leq q$, and we obtain the following result:

$$\max_{i,j} |C_{i,j}| \leq \frac{1}{p}(\min_k \mu_k)^{-1/p} = \frac{1}{p}\|A^{-1/p}\|_2.$$

Finally, we compute the Frobenius norm of $\Delta_1$:

$$\begin{aligned}
\|\Delta_1\|_F^2 &= \sum_{i,j} \left|[\hat{\Delta}_1]_{ij}\right|^2 = \sum_{i,j} \left|C_{ij}[\hat{\tilde{E}}]_{ij}\right|^2 \\
&\leq \left(\max_{i,j} |C_{ij}|\right)^2 \sum_{i,j} \left|[\hat{\tilde{E}}]_{ij}\right|^2 \\
&= \left(\frac{1}{p}\|A^{-1/p}\|_2\right)^2 \|\hat{\tilde{E}}\|_F^2.
\end{aligned}$$

Taking the square root and using $\|\hat{\tilde{E}}\|_F = \|\tilde{E}\|_F = \|A^{-1/2}EA^{-1/2}\|_F$, we obtain:

$$\|\Delta_1\|_F \leq \frac{1}{p}\|A^{-1/p}\|_2 \cdot \|A^{-1/2}EA^{-1/2}\|_F \leq \frac{1}{p}\|A^{-1/p}\|_F \cdot \|A^{-1/2}EA^{-1/2}\|_F.$$

Hence, we obtain the desired results. □

## I. Further Discussion of the Decision Criterion of Eschenhagen et al. (2026) and Ours

This section clarifies a key distinction between the quantity controlled by the criterion of Eschenhagen et al. (2026) and the quantity that directly governs the stability of Shampoo-style updates. Their criterion measures how well a stale eigenbasis diagonalizes the *current Kronecker statistic*, or equivalently, how accurately the matrix itself is approximated in that stale basis. By contrast, the Shampoo update depends on the *inverse-root operator*

$$F_\epsilon(X) := (X + \epsilon\mathbf{I})^{-1/p},$$

and hence the relevant quantity is not the matrix approximation error itself, but the induced *relative inverse-root operator error*. These two notions need not agree. In particular, a basis may look nearly diagonal under a residual criterion while still inducing a substantial error after the nonlinear inverse-root map is applied.

To make this distinction precise, we first provide a simple counterexample showing that the diagonalization residual may vanish while the inverse-root operator error remains $\Theta(1)$. We then explain the mechanism via first-order perturbation theory for matrix functions, which reveals that (i) eigenvalue drift and (ii) eigenspace mixing enter the inverse-root map in fundamentally different, anisotropic ways. Finally, we revisit the "warm-started QR" interpretation of Eschenhagen et al. (2026) and argue that, under their trigger policy, the QR refinement is often invoked precisely in a regime that is operationally closer to a *cold start* than to a genuinely local warm-start refinement.

**What their criterion controls, and what ours controls.**   Let $\widehat{Q}$ be a stale eigenbasis and define the stale-basis diagonal approximation

$$\widehat{L}_t := \widehat{Q}\,\mathrm{diag}(\widehat{Q}^\top L_t\widehat{Q})\,\widehat{Q}^\top.$$

The criterion of Eschenhagen et al. (2026) controls the relative approximation error

$$\frac{\|L_t - \widehat{L}_t\|_F}{\|L_t\|_F} = \frac{\|\widehat{Q}^\top L_t\widehat{Q} - \mathrm{diag}(\widehat{Q}^\top L_t\widehat{Q})\|_F}{\|\widehat{Q}^\top L_t\widehat{Q}\|_F},$$

that is, the off-diagonal mass of the current statistic in the stale basis. This is a meaningful quantity for assessing whether $\widehat{Q}$ approximately diagonalizes $L_t$. However, the Shampoo update is driven not by $L_t$ itself, but by $(L_t + \epsilon\mathbf{I}_m)^{-1/p}$. Therefore, from the viewpoint of update accuracy and numerical stability, the more relevant quantity is the *relative inverse-root operator error*

$$\frac{\|F_\epsilon(L_t) - F_\epsilon(\widehat{L}_t)\|_F}{\|F_\epsilon(\widehat{L}_t)\|_F}.$$

Our proxy is designed to approximate this latter quantity directly, rather than the former.

**A counterexample: small diagonalization residual but large inverse-root operator error.**   Consider the $2 \times 2$ SPD matrix

$$L_\delta = \begin{bmatrix} \delta & c\sqrt{\delta} \\ c\sqrt{\delta} & 1 \end{bmatrix}, \qquad 0 < c < 1,$$

with $\delta > 0$ sufficiently small. The condition $0 < c < 1$ ensures $L_\delta \succ 0$ for all sufficiently small $\delta$. Suppose the stale eigenbasis is the standard basis, i.e. $\widehat{Q} = \mathbf{I}$. Then the diagonalization residual used by Eschenhagen et al. (2026) is

$$\frac{\|L_\delta - \mathrm{diag}(L_\delta)\|_F}{\|\mathrm{diag}(L_\delta)\|_F} = \frac{\sqrt{2}\,c\sqrt{\delta}}{\sqrt{\delta^2 + 1}} \xrightarrow[\delta \to 0]{} 0.$$

Hence, from the viewpoint of diagonalization residual, the stale basis appears asymptotically "accurate."

However, the inverse-root operator error behaves very differently. For $p = 2$, compare the exact inverse root with the diagonal approximation in the same basis:

$$\frac{\|L_\delta^{-1/2} - \mathrm{diag}(L_\delta)^{-1/2}\|_F}{\|\mathrm{diag}(L_\delta)^{-1/2}\|_F} \xrightarrow[\delta \to 0]{} \frac{1}{\sqrt{1 - c^2}} - 1.$$

In particular, this limit is strictly positive for every fixed $c \in (0, 1)$. Thus, although the off-diagonal entry $c\sqrt{\delta}$ vanishes, the relative inverse-root error does *not* vanish. Equivalently, the diagonalization residual can be arbitrarily small while the operator error relevant to the Shampoo update remains $\Theta(1)$.

The reason is transparent from the smallest eigenvalue. A direct calculation shows

$$\lambda_{\min}(L_\delta) = \frac{\delta + 1 - \sqrt{(1-\delta)^2 + 4c^2\delta}}{2} = (1 - c^2)\delta + O(\delta^2), \qquad \delta \downarrow 0.$$

Hence the low-spectrum direction becomes increasingly ill-conditioned, and the inverse-root map amplifies perturbations in precisely that direction. The diagonalization residual does not capture this effect.

**Why this happens: the inverse-root map is anisotropic.** The above phenomenon is not peculiar to $2 \times 2$ matrices; it reflects the general anisotropy of matrix inverse-root maps. Let

$$X = Q\Lambda Q^\top, \quad \Lambda = \mathrm{diag}(\lambda_1, \ldots, \lambda_n),$$

and let $\Delta X$ be a symmetric perturbation. By the Daleckii–Kreĭn formula (Lemma C.11) for matrix functions, the first-order variation is

$$F_\epsilon(X + \Delta X) - F_\epsilon(X) \approx DF_\epsilon(X)[\Delta X] = Q\big(G \odot (Q^\top \Delta X Q)\big)Q^\top,$$

where $\odot$ is the Hadamard product and $G$ is the divided-difference matrix

$$G_{ij} = \begin{cases} \dfrac{f_\epsilon(\lambda_i) - f_\epsilon(\lambda_j)}{\lambda_i - \lambda_j}, & i \neq j, \\ f'_\epsilon(\lambda_i), & i = j, \end{cases} \qquad f_\epsilon(\lambda) = (\lambda + \epsilon)^{-1/p}.$$

This representation makes two facts explicit.

*(i) Diagonal drift can dominate even when the off-diagonal residual is tiny.* The diagonal entries $(Q^\top \Delta X Q)_{ii}$ are weighted by

$$|f'_\epsilon(\lambda_i)| = \frac{1}{p}(\lambda_i + \epsilon)^{-(p+1)/p},$$

which is the largest near the smallest eigenvalues. Therefore, even when the stale basis nearly diagonalizes $X$ in the usual residual sense, small diagonal drift in low-spectrum directions can induce a large change in the inverse-root operator. The counterexample above is exactly of this type.

*(ii) Off-diagonal mass is not uniformly important.* A diagonalization residual treats all off-diagonal entries of $\widehat{Q}^\top X \widehat{Q}$ on roughly equal footing. In contrast, the inverse-root response depends on *where* those off-diagonal components occur through the divided differences $G_{ij}$. Mixing between well-conditioned high-eigenvalue directions may be relatively harmless, whereas perturbations that act on, or mix into, low-spectrum directions can be strongly amplified. Thus, the inverse-root map is intrinsically anisotropic, and a uniform off-diagonal residual is not aligned with the actual update sensitivity.

**Implication for decision rules.** The preceding discussion shows that the question

$$\text{``How diagonal does } \widehat{Q}^\top L_t \widehat{Q} \text{ look?''}$$

is not the same as

$$\text{``How accurate is } (L_t + \epsilon \mathbf{I}_m)^{-1/p} \text{ under the stale approximation induced by } \widehat{Q}\text{?''}$$

The former concerns matrix approximation in a stale eigenspace; the latter concerns the fidelity of a nonlinear, spectrum-sensitive operator that directly enters the update. Accordingly, a decision rule based only on diagonalization residual may be poorly calibrated for Shampoo-style stability control.

For this reason, we do not use the diagonalization-residual trigger of Eschenhagen et al. (2026) as the principal control signal. Instead, we target the relative inverse-root operator error itself. Concretely, our sensing objective is the relative gap

$$\frac{\|\Delta_L(t)\|_F}{\|\widehat{L}_t^{-1/p}\|_F}, \qquad \Delta_L(t) := (L_t + \epsilon_t \mathbf{I}_m)^{-1/p} - \widehat{L}_t^{-1/p},$$

and our proxy is derived from a first-order approximation to this quantity in the stale eigenspace. In particular, it reweights the drift by inverse spectral factors and therefore tracks precisely the low-spectrum sensitivity that the diagonalization residual misses.

**On the (mis)use of warm-started QR: a cold-started schedule in disguise.** The idea of refining a stale basis by a small number of QR iterations is natural. Our criticism is not of warm-started QR *per se*, but of the regime in which it is invoked under the trigger policy of Eschenhagen et al. (2026). Their QR refinement is activated precisely when the residual criterion fails, i.e. when the current basis is already a poor diagonalizer of the current statistic. In that regime, the iterate is no longer necessarily close to the desired invariant subspace, so the favorable local behavior associated with a genuine warm start cannot be assumed. Operationally, the refinement is therefore often initiated in what is better described as a *cold-started regime*.

This matters computationally. Dense QR iteration costs $\mathcal{O}(n^3)$ per step (and similarly $\mathcal{O}(m^3)$ for the other factor), and several iterations may be required before entering an accurate regime. In contrast, a direct eigensolver such as `eigh` performs the standard reduction once and then solves the resulting tridiagonal problem efficiently. Hence, when the trigger calls QR only after the basis has already drifted substantially, the cumulative cost of multiple dense QR iterations can be comparable to, or exceed, that of direct eigendecomposition.

This interpretation is also consistent with the empirical findings reported by Eschenhagen et al. (2026): on their Imagewoof ViT experiment, adaptive QR was slower in wall-clock time than adaptive `eigh`, and even the default one-step SOAP-style refinement was slightly slower and reached a worse final loss. Indeed, they ultimately emphasize using the criterion mainly to decide when to call `eigh`, rather than to justify QR refinement as the primary computational primitive. From our viewpoint, this is exactly what one would expect when the trigger is not aligned with the inverse-root operator error and when the refinement is invoked only after the stale basis has already become substantially inaccurate.

### I.1. Real-task complement: cost of residual-based QR refinement

We examine a complementary practical question on real trigger states collected during ViT training: how expensive is it to satisfy the diagonalization-residual criterion by QR refinement? Each saved state contains a stale basis together with the corresponding current Kronecker factor at a refresh-check step, and all runtimes are normalized by the direct `EVD` time measured on the same state and implementation path.

Table 1 reports the median number of QR iterations required to satisfy the residual threshold $\tau$ in Eschenhagen et al. (2026) for these saved trigger states.

| Model | Residual threshold ($\tau$) | QR iters | Median runtime / direct `EVD` | Median residual after QR |
|-------|------------------------------|----------|-------------------------------|--------------------------|
| ViT   | 0.1  | 1 | 0.52× | 0.695 |
| ViT   | 0.1  | 2 | 1.01× | 0.451 |
| ViT   | 0.1  | 4 | 1.97× | 0.230 |
| ViT   | 0.1  | 8 | 3.93× | 0.098 |
| ViT   | 0.25 | 1 | 0.36× | 0.682 |
| ViT   | 0.25 | 2 | 0.65× | 0.360 |
| ViT   | 0.25 | 4 | 1.27× | 0.125 |
| ViT   | 0.25 | 8 | 2.55× | 0.060 |

*Table 1.* QR refinement cost to satisfy the diagonalization-residual criterion on saved ViT trigger states.

The median residual drops below $\tau = 0.1$ (or $\tau = 0.25$) only after 8 (or 4) QR iterations, at which point the median runtime is already $3.93\times$ (or $1.27\times$) that of direct `EVD`. Thus, for real saved ViT trigger states, a single QR step is typically insufficient to satisfy the residual criterion, whereas sufficient QR refinement to satisfy it is already slower than direct `EVD`.

*Table 2.* Hyperparameters

| *(a)* ViT on ImageNet. | | *(b)* Conformer on LibriSpeech. | | *(c)* GPT-2 on WikiText. | |
|---|---|---|---|---|---|
| **Hyperparameter** | **Value** | **Hyperparameter** | **Value** | **Hyperparameter** | **Value** |
| Epoch | 90 | Epoch | 90 | Epoch | 30 |
| Batch size | 1024 | Batch size | 1024 | Batch size | 512 |
| Momentum | (0.95, 0.995) | Momentum | (0.95, 0.995) | Momentum | (0.95, 0.99) |
| WD | $4.2 \times 10^{-4}$ | WD | $5.0 \times 10^{-4}$ | WD | $1.0 \times 10^{-2}$ |
| LR | $\{1.75, 2.00, 2.20, 2.35\} \times 10^{-3}$ | LR | $\{1.75, 2.00, 2.25\} \times 10^{-3}$ | LR | $\{1.0, 3.5, 6.0\} \times 10^{-3}$ |
| LR scheduling | Warm-up (5%) + cosine decay | LR scheduling | Warm-up (5%) + cosine decay | LR scheduling | Warm-up (5%) + cosine decay |
| LR grafting | Adam | LR grafting | Adam | LR grafting | Adam |

## J. Further Empirical Results

In this section, we discuss additional empirical results.

### J.1. Experimental Settings

We implement FOAM and all baselines in PyTorch 2.8 with CUDA 12.8. For the ViT (ImageNet-1k) task, we used the NVIDIA RTX A6000 × 4 units; as an exception, we used NVIDIA RTX Pro 6000 × 4 units only for the experiment shown in Figure 4. Next, we used NVIDIA RTX Pro 6000 × 4 units for the Conformer (Librispeech) task, and we used NVIDIA DGX B200 × 2 units for the language model task. Unless stated otherwise, we use an identical batch size, momentum coefficients, step size, and its scheduling, and weight decay (WD) for AdamW, Stale Shampoo[3], FOAM, DR-Shampoo (Eschenhagen et al. (2026)), and SOAP [4] (Table 2a–2c). We couldn't find the source code for Eschenhagen et al. (2026); hence, we implemented it.

For the learning rate (LR), we selected a single LR using only the Stale Shampoo setting (from the candidate grid in the tables) and then compared baseline optimizers and FOAM under this LR. We apply Adam LR grafting (Shi et al., 2023) for all experiments except AdamW and SOAP. Additional baselines and FOAM specific hyperparameters (*e.g.,* $\mathbf{f}$, $\tau$, $\epsilon_{\max}$) are reported in each Figure.

### J.2. ViT on ImageNet: More Results

*Table 3.* Comparison of wall-clock time and training loss on the ViT task.

| Optimizer | $(\mathbf{f}, \tau, \epsilon_0, \epsilon_{\max})$ | Train Loss | Validation accuracy | Wall-clock Time |
|---|---|---|---|---|
| AdamW | (N/A, N/A, $10^{-9}$, N/A) | 0.635 | 72.39 % | 1,258 (minute) |
| stale Shampoo | (20, N/A, $10^{-9}$, N/A) | 0.498 | 75.13% | 1,415 (minute) |
| DR-Shampoo | (20, 0.75, $10^{-9}$, N/A) | 0.531 | 73.56 % | 1,310 (minute) |
| FOAM (Ours) | (20, 0.75, $10^{-9}$, $3 \times 10^{-7}$) | 0.458 | 75.72 % | 1,300 (minute) |
| SOAP | (20, N/A, $10^{-9}$, N/A) | 0.485 | 75.1% | 1,360 (minute) |

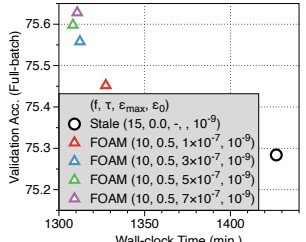 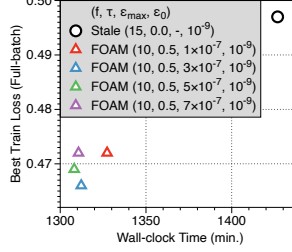 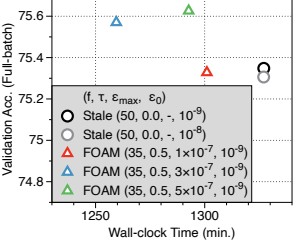 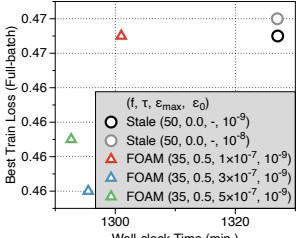

*Figure 6.* ViT: Best training loss and validation accuracy

Our experimental results in Table 3 show that our algorithm FOAM outperforms several baseline optimizers. Next, Figure 6 shows the validation accuracy and training loss as functions of wall-clock time (minutes). In general, FOAM exhibits a wall-clock advantage without loss of performance. This demonstrates that reducing the eigendecomposition and adjusting

---

[3] https://github.com/facebookresearch/optimizers.git
[4] https://github.com/nikhilvyas/SOAP.git

*Table 4.* Proxy ($h_t$) computation cost and `EVD` cost under different settings.

| Model | Setting | Proxy computation cost | EVD cost |
|---|---|---|---|
| ViT | ($\tau = 0.5$; $\mathbf{f} = 25$; $\epsilon_{\max} = 3 \times 10^{-7}$) | $\sim 4.72\%$ | $\sim 17.67\%$ |
| ViT | ($\tau = 0.1$; $\mathbf{f} = 50$; $\epsilon_{\max} = 2 \times 10^{-7}$) | $\sim 2.10\%$ | $\sim 29.48\%$ |

*Table 5.* Dimension-wise profiling results for ViT.

| Model & Setting | Dimensions | profiling |
|---|---|---|
| ViT ($\tau = 0.5$; $\mathbf{f} = 25$; $\epsilon_{\max} = 3 \times 10^{-7}$) | 384 | 0.256 ms |
| | 512 | 0.264 ms |
| | 1024 | 0.286 ms |

epsilon via the error proxy effectively controls both wall-clock time gains and staleness-induced preconditioner errors.

**Computational Overhead analysis** In Table 4, we measured proxy-computation time and `EVD` time separately and reported each as a percentage of the total optimizer time over the full run.

In Table 5, we also recorded (Kronecker) factor-level timings by matrix dimensions, which showed that across ViT factor sizes (384-1024), proxy computation remained nearly flat and was much smaller than the `EVD` cost. Thus, when comparing the corresponding median EVD times of 13.29 ms, 18.44 ms, and 14.98 ms, the proxy remained roughly $50 - 70\times$ cheaper than `EVD` across the real-factor dimensions encountered during training.

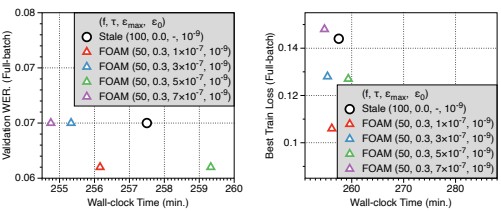

*Figure 7.* Conformer: Best train. loss and valid. WER

| Optimizer | ($\mathbf{f}, \tau, \epsilon_0, \epsilon_{\max}$) | Train Loss | Validation WER | Wall-clock Time |
|---|---|---|---|---|
| AdamW | (N/A, N/A, $10^{-9}$, N/A) | 0.42 | 0.099 | 205 (minute) |
| stale Shampoo | (50, N/A, $10^{-9}$, N/A) | 0.17 | 0.069 | 264 (minute) |
| DR-Shampoo | (50, 0.4, $10^{-9}$, N/A) | 0.18 | 0.065 | 294 (minute) |
| FOAM (Ours) | (50, 0.4, $10^{-9}$, $3 \times 10^{-7}$) | 0.14 | 0.065 | 257 (minute) |
| SOAP | (50, N/A, $10^{-9}$, N/A) | 0.17 | 0.067 | 265 (minute) |

*Table 6.* Comparison of wall-clock time and training loss on the Conformer task.

## J.3. Conformer on LibriSpeech: More Results

In Figure 6, we show the superior performance of `FOAM` compared to several baseline optimizers. Our experimental results in Figure 7 show the validation WER and training loss as functions of wall-clock time (minutes). In general, `FOAM` exhibits a wall-clock advantage without loss of performance. However, the `FOAM` $(50, 0.3, 7 \times 10^{-7}, 10^{-9})$ setting exhibits a higher training loss than Stale Shampoo, indicating that excessive damping degrades the preconditioning eff

*Table 7.* Comparison of wall-clock time and training loss on the language model task.

| Optimizer | ($\mathbf{f}, \tau, \epsilon_0, \epsilon_{\max}$) | Train Loss | Val PPL | Wall-clock Time |
|---|---|---|---|---|
| AdamW | (N/A, N/A, $10^{-12}$, N/A) | 2.30 | 18.37 | 609 (minute) |
| stale Shampoo | (100, N/A, $10^{-12}$, N/A) | 2.31 | 18.26 | 630 (minute) |
| FOAM | (75, 0.4, $10^{-12}$, $1 \times 10^{-10}$) | 2.29 | 18.22 | 622 (minute) |
| FOAM | (75, 0.4, $10^{-12}$, $2 \times 10^{-10}$) | 2.30 | 18.22 | 621 (minute) |
| FOAM | (75, 0.4, $10^{-12}$, $3 \times 10^{-10}$) | 2.30 | 18.27 | 625 (minute) |
| DR-Shampoo | (75, 0.4, $10^{-12}$, N/A) | 2.31 | 18.33 | 623 (minute) |
| SOAP | (75, N/A, $10^{-12}$, N/A) | 2.31 | 18.33 | 642 (minute) |

*Table 8.* Comparison of wall-clock time and training loss on the ViT task with DR-Shampoo and Ours.

| Optimizer | ($\mathbf{f}, \tau, \epsilon_0, \epsilon_{\max}$) | Train Loss | Validation accuracy | Wall-clock Time |
|---|---|---|---|---|
| DR-Shampoo | (20, 0.25, $10^{-9}$, $-$) | 0.489 | 74.7% | 1,354 (minute) |
| FOAM | (20, 0.25, $10^{-9}$, $3 \times 10^{-7}$) | 0.468 | 75.38% | 1,310 (minute) |
| DR-Shampoo | (20, 0.5, $10^{-9}$, $-$) | 0.494 | 74.2% | 1,326 (minute) |
| FOAM | (20, 0.5, $10^{-9}$, $3 \times 10^{-7}$) | 0.467 | 75.6% | 1,300 (minute) |
| DR-Shampoo | (20, 0.75, $10^{-9}$, $-$) | 0.531 | 73.56% | 1,310 (minute) |
| FOAM | (20, 0.75, $10^{-9}$, $3 \times 10^{-7}$) | 0.458 | 75.72% | 1,300 (minute) |

## J.4. Language model on Wikitext-103-v1

We conducted experiments with the GPT-2 base (247M) model and confirmed `FOAM`'s robust effectiveness and superior efficiency (see Table 7).

## J.5. Compare with Eschenhagen et al. (2026)

For the diagonal-residual criterion method (Eschenhagen et al., 2026), we refresh using EVD rather than QR iterations. Our reason is that, as observed in Table 1, the number of QR iterations needed to satisfy the residual criterion was already comparable to, or slower than, direct EVD. Furthermore, although Eschenhagen et al. (2026) proposed adaptive QR, they also observed in their experiments that the QR algorithm was less efficient than EVD in terms of wall-clock time and explicitly stated that they used EVD when the diagonal-residual was greater than the threshold $\tau$ (refer to subsection 4.2 in Eschenhagen et al. (2026)). Taking these factors into account, we conducted experiments using EVD rather than warm-started QR iteration. We report additional experimental results in Table 8.

## K. Proxy Validation Experiments

### K.1. Synthetic example

**Goal.** We test whether the surrogate

$$h(\epsilon) = RC(\epsilon)\frac{\alpha(\epsilon)}{p}$$

tracks the ground-truth relative inverse-root operator error and yields a useful trigger signal for eigendecomposition.

**Data generation.** For each configuration, we sample an SPD matrix

$$A = Q\,\text{diag}(\lambda)Q^\top, \qquad \lambda_i = i^{-r},$$

where $Q$ is obtained by QR factorization of a Gaussian matrix. To model controlled stale drift, we use a PSD perturbation

$$E = BB^\top \succeq 0,$$

rescaled as

$$E \leftarrow \frac{E}{\|E\|_F}\big(\texttt{E\_scale} \cdot \|A\|_F\big), \qquad A_{\text{new}} = A + E.$$

We use PSD perturbations to preserve positive definiteness uniformly across the sweep.

**Damping sweep and operators.** We sweep $\epsilon$ on a logarithmic grid

$$\epsilon \in [\epsilon_{\min}, \epsilon_{\max}]$$

with $K = \texttt{eps\_steps}$ points. For inverse-root order $p$, define the stale and fresh damped inverse-root operators by

$$P_s(\epsilon) = Q\,\text{diag}\big((\lambda + \epsilon)^{-1/p}\big)Q^\top, \qquad P_f(\epsilon) = U\,\text{diag}\big((\tilde{\lambda} + \epsilon)^{-1/p}\big)U^\top,$$

where

$$(\tilde{\lambda}, U) = \texttt{EVD}(A_{\text{new}}).$$

**Ground-truth error.** We measure the target relative operator error by

$$\Delta(\epsilon) = \frac{\|P_f(\epsilon) - P_s(\epsilon)\|_F}{\|P_s(\epsilon)\|_F}.$$

**Proxy and baseline.** Let

$$\hat{E} = Q^\top E Q, \qquad D = \text{diag}(\lambda + \epsilon).$$

We compute

$$RC(\epsilon) = \|D^{-1/2}\hat{E}D^{-1/2}\|_F, \qquad \alpha(\epsilon) = \frac{\|(\lambda + \epsilon)^{-1/p}\|_\infty}{\|(\lambda + \epsilon)^{-1/p}\|_2}, \qquad h(\epsilon) = RC(\epsilon)\frac{\alpha(\epsilon)}{p}.$$

As a baseline, we use a *damped* diagonalization-residual score in the stale basis:

$$L(\epsilon) = U\,\text{diag}(\tilde{\lambda} + \epsilon)U^\top, \qquad \hat{L}(\epsilon) = Q^\top L(\epsilon)Q, \qquad d(\epsilon) = \frac{\|\hat{L}(\epsilon) - \text{diag}(\hat{L}(\epsilon))\|_F}{\|\hat{L}(\epsilon)\|_F}.$$

We include the same damping level $\epsilon$ in $d(\epsilon)$ so that the proxy and the baseline are compared under a matched damped operator scale.

**Evaluation.** For each configuration, we compute Pearson and Spearman correlations between $\log_{10}\Delta(\epsilon)$ and $\log_{10}h(\epsilon)$ across the $K = 25$ values of $\epsilon$, and we compute the ROC-AUC for detecting "refresh-needed" samples defined as the top 20% by $\Delta(\epsilon)$ within that configuration. We then summarize these configuration-wise statistics by their median and interquartile range (IQR) across all configurations. For calibration and alignment, we additionally pool all samples across all configurations and examine (i) the empirical distribution of $\Delta(\epsilon)/h(\epsilon)$ and (ii) the log-scale scatter of $\log_{10}h(\epsilon)$ versus $\log_{10}\Delta(\epsilon)$.

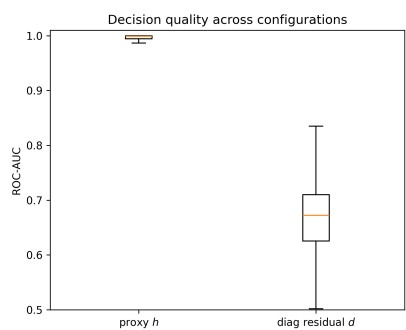

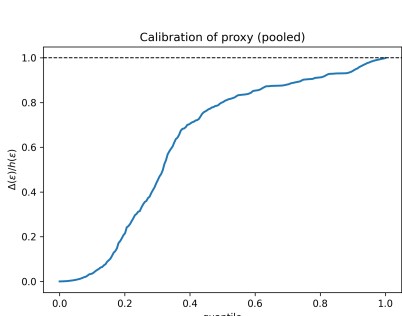

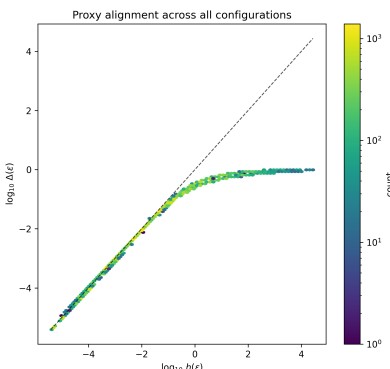

*(a)* **Decision utility (configuration-wise).** Distribution across the 150 configurations of ROC-AUC for identifying the top-20% samples by $\Delta(\epsilon)$.

*(b)* **Calibration (pooled).** Empirical quantiles of $\Delta(\epsilon)/h(\epsilon)$ over all 56,250 samples; values below 1 indicate conservativeness.

*(c)* **Alignment (pooled).** Hexbin of $\log_{10} h(\epsilon)$ versus $\log_{10} \Delta(\epsilon)$ over all 56,250 samples, with $y = x$ shown as a reference line.

*Figure 8.* **Synthetic validation of the proxy.** Panel (a) reports configuration-wise decision quality, showing that $h(\epsilon)$ provides an almost perfect trigger signal for eigendecomposition and substantially outperforms the diagonalization-residual baseline $d(\epsilon)$. Panels (b) and (c) pool all samples across the sweep: the calibration plot shows that $\Delta(\epsilon)/h(\epsilon) \leq 1$ throughout the evaluated range, indicating conservative behavior in this experiment, while the alignment plot shows strong monotone agreement between $h(\epsilon)$ and the ground-truth relative operator error $\Delta(\epsilon)$.

**Sweep.** We sweep

$$d \in \{256, 512, 1024\}, \qquad p \in \{2, 4\}, \qquad r \in \{0.5, 1.0, 1.5, 2.0, 2.5\},$$

$$\texttt{E\_scale} \in \{10^{-5}, 10^{-4}, 10^{-3}, 10^{-2}, 10^{-1}\}, \qquad \epsilon \in [10^{-8}, 10^{-2}]$$

(log-spaced), with $K = 25$ damping values and $T = 15$ trials per configuration. This yields 150 configurations and 56,250 total samples.

**Findings.** Across the 150 configurations, the proposed proxy $h(\epsilon)$ provides near-perfect decision utility: the configuration-wise ROC-AUC has median 0.9988 (IQR [0.9944, 1.000]; worst-case 0.902), substantially outperforming the diagonalization-residual baseline $d(\epsilon)$ (median 0.672, IQR [0.625, 0.710]). The configuration-wise log-scale alignment is also strong, with median Pearson correlation 0.9994 and median Spearman correlation 0.9982. For pooled calibration over all 56,250 samples, the ratio $\Delta(\epsilon)/h(\epsilon)$ has median 0.761 and maximum 0.999, indicating that, throughout our sweep, the proxy remains conservative while staying tightly aligned with the target relative error; see Figure 8.

