# OpenReview forum: "FOAM: Frequency and Operator-Error Based Adaptive Damping Method for Reducing Staleness-Oriented Error for Shampoo"
_ICML.cc/2026/Conference — ICML 2026 regular_

### Official Review · Reviewer_LD8P · 2026-03-11

**Soundness:** 3
**Presentation:** 3
**Significance:** 2
**Originality:** 3
**Overall Recommendation:** 4
**Confidence:** 3

**Summary:**

This paper proposes FOAM, a Shampoo-style preconditioned optimizer that uses stale error statistics together with an adaptive mechanism intended to stabilize and improve stale preconditioning. The motivation is practical: full Shampoo-style updates are expensive, and stale updates are attractive computationally but can degrade optimization if the preconditioner becomes inaccurate. The paper provides a regret-style theoretical analysis and empirical results intended to show that the proposed method improves over stale Shampoo.

**Compliance With Llm Reviewing Policy:**

Affirmed.

**Final Justification:**

The rebuttal improved my assessment, hence I stick to my improved score.

**Key Questions For Authors:**

1) The main theoretical result appears to rely on convexity. Can the authors reframe so that this is explicit from the start, and explain precisely how the regret theorem should be interpreted relative to the nonconvex experimental setting? If there is any direct convergence implication (or not) the authors intend the reader to draw, this should be stated explicitly.

2) Theorem 5.4 seems to assume
\begin{equation*}
\max_t |W_t - W^\star| = D \quad \text{for all } W^\star.
\end{equation*}
As written this is impossible. What is the intended assumption?

3) The analysis seems to rely on a fixed step size. Is this assumption essential to the proof, or only a simplifying choice? Please clarify how the theory relates to the actual optimizer configuration used in the experiments.

4) How was stale Shampoo tuned in the experiments? Were stale frequency and damping swept with similar care as for FOAM?

5) Why were there no comparisons against other nearby preconditioned optimizers discussed in related work, and why were there no experiments in a moderately large LLM setting? A convincing answer here could improve my assessment of significance.

**Limitations:**

No. The paper does acknowledge some limitations, in particular that the current theory is convex-only, which is good. However the limitations discussion should be expanded to better address the theory-practice gap: the main theorem is a discounted-regret result rather than a direct nonconvex convergence guarantee, and the empirical evaluation is limited relative to the practical scope suggested by the paper. I do not see major negative societal-impact issues beyond the usual energy and compute considerations of large-scale training, but the methodological limitations should be discussed more candidly.

**Strengths And Weaknesses:**

The paper addresses a practically relevant problem. Second-order or preconditioned methods are often limited by the cost of updating the preconditioner, so improving stale-update variants is a meaningful direction. The proposed method is also reasonably well motivated at a high level, and the empirical section suggests that the method can outperform a stale Shampoo baseline in the tested settings. I also appreciate that the paper attempts to provide theory rather than only empirical evidence.

My main concerns are about soundness and clarity of the theoretical claims. The theory appears to be framed too broadly in parts of the paper: the main regret result is in the convex setting, but the paper does not make this limitation sufficiently explicit early enough. Since the experiments are in the modern deep learning regime, the gap between the convex regret analysis and the practical setting should be stated more clearly. The paper also uses regret analysis, but does not explain what the regret bound actually implies for optimization behavior or convergence. As written, the reader is left unclear on whether the theorem should be interpreted as a convergence guarantee, a stability result, or mainly a qualitative justification for stale preconditioning.

I also found at least two (minor) technical presentation issues that are not minor. First, Theorem 5.4 appears to contain an impossible assumption of the form $ \max_t |W_t - W^\star| = D $ for all comparators $W^\star$, which cannot be correct as stated. Presumably this is meant either for a fixed comparator. Second, the pseudocode in Algorithm 1 may contain an indexing inconsistency: it appears to compute a refreshed preconditioner at time $t+1$ but apply the $t$-th one in the update, and the refresh schedule in the algorithm and appendix is not obviously aligned.

A further issue is the role of the step size in the theory. The analysis appears to rely on a fixed step size, while practical methods are typically trained with tuned schedules. The paper should clarify whether the fixed step size assumption is essential for the argument or merely a proof convenience, and explain how the theorem should be interpreted relative to the optimizer configuration used in practice.

The empirical evaluation is promising but too narrow for the strength of the claims. I would have liked to see stronger baseline coverage, especially since the related-work section mentions several nearby preconditioned methods from the same family. It is also unclear whether the stale Shampoo baseline received a comparably careful sweep over its key hyperparameters, in particular frequency and damping. My understanding is that methods in this family are often motivated by large-scale language-model training, so the absence of any moderately large LLM experiment weakens the practical significance claim.

On originality, I view the contribution as moderately novel. The paper does not introduce an entirely new optimization paradigm, but it does propose a concrete modification of stale Shampoo together with an accompanying analysis. That is a legitimate contribution, but in my view the current version does not yet provide a strong claim of significance.

---

> ### Author Rebuttal · Authors · 2026-03-31
>
> We thank the reviewer for the constructive feedback and for recognizing the practical relevance and meaningful direction of our work. We appreciate the acknowledgment of our motivation, theoretical novelty, and the proposed method's ability to outperform the baselines. Below are our responses to the questions. We are ready to provide any further clarification immediately.
>
> ---
>
> **1. Regarding the convexity assumption, which does not disrupt the flow of our structure.**
>
> Our primary goal was to bound the linear loss (written in Appendix G, step 2 $(1-\beta)(\sum_{t=1}^T \beta^{T-t} < G_t, W_t - W^*>$) and the convexity assumption was specifically introduced to interpret this bound as discounted regret. This follows the standard analytical framework of [1, 2], allowing us to express the objective relative to a fixed comparator for clearer theoretical insight.
>
> We clarify that Theorem 5.4 is not intended as a non-convex convergence guarantee. Rather, its role is qualitative and mechanistic: it explains how stale conditioning creates a penalty in a convergence-related objective and why increasing damping counteracts it.
>
> In the revised manuscript, we will explicitly state these scope limitations in the abstract, introduction, and theorem discussion to prevent any misinterpretation of Theorem 5.4 as a direct proof for non-convex settings.
>
> [1] Understanding Adam Optimizer via Online Learning of Updates: Adam is FTRL in Disguise, ICML’24
>
> [2] Optimal Stochastic Non-smooth Non-convex Optimization through Online-to-Non-convex Conversion, ICML’23
>
> ---
>
> **2. Clarification of claims and notation.**
>
> (a) Clarification of the term $D$ and comparator dependency.
> The term $D$ is intended to be comparator-dependent, following the standard convention used in Shampoo-style analyses. More precisely, for any fixed comparator $(W^\*\)$, we define
> $D(W^\*) := \max_t \|W_t - W^\*\|_F.$
> We will revise the theorem statement and the appendix discussion to explicitly reflect this definition. This correction clarifies the quantification but does not change the substance of the proof.
>
> (b) time Indexing clarity.
> The update rule $\mathcal{U}(\cdot)$ was modularized specifically to enable smart control of resource reuse, which led to the current indexing inconsistencies in the pseudocode. We will revise the algorithms to explicitly clarify the timing of preconditioner refreshes. Furthermore, we will ensure full consistency in time notation between the algorithm and the Appendix to remove any ambiguity regarding staleness. These clarifications will be strictly reflected in the revised manuscript.
>
> ---
>
> **3. On the fixed-step-size assumption.**
>
> As the reviewer noted, the fixed step size was a choice for mathematical simplicity to isolate the core theoretical mechanisms. We will explicitly clarify this distinction in the revised manuscript. However, we would like to emphasize that FOAM remains highly robust in practice, even with a varying step-size schedule. The consistency between our theory and the empirical results suggests that the fundamental advantages of FOAM are not strictly dependent on a fixed step size.
>
> ---
>
> **4. Hyperparameter tuning**
>
> We would like to clarify that stale shampoo was thoroughly tuned (specifically $\mathbf{f}$ and $\epsilon$) to establish a strong baseline.
> Rather than reporting only FOAM’s single best configuration, we reported a wide range of its settings to demonstrate its robustness. As seen in Figure 1 of the manuscript, the majority of FOAM’s configurations consistently outperform the optimized Shampoo baseline without requiring task-specific tuning. This confirms that FOAM provides a ‘plug-and-play’ advantage across various setups. We detailed our hyperparameter search space below and will include this in Appendix J of the revision to ensure transparency.
>
> ---
>
> **5. LLM experiment.**
>
> Please refer to our response to **Question 3 from Reviewer NS41** for details on the GPT-2 small experiments. In short, our GPT-2 experiments demonstrate that our method outperforms SOAP, its recent variant, and Stale Shampoo in both performance and wall-clock time.

---

> > ### Author Rebuttal · Reviewer_LD8P · 2026-03-31
> >
> > The rebuttal improved my assessment. In particular, the added experiments and a clearer discussion of FOAM’s weaknesses make the empirical contribution more convincing. I still have some concerns about the framing and interpretation of the theory, however I now view these as issues that should be fixed in revision rather than reasons to reject the paper.

---

> > > ### Author Response · Authors · 2026-04-06
> > >
> > > We sincerely appreciate that our rebuttal has satisfactorily resolved your concerns.
> > >
> > > In the revised manuscript, we will explain more clearly the meaning, scope, and intended interpretation of the theoretical results, and make explicit that they are meant to motivate and explain FOAM’s mechanism rather than serve as a direct nonconvex convergence guarantee. We will also further clarify FOAM’s limitations and discuss how these limitations suggest meaningful directions for future research.
> > >
> > > Thank you again for your careful and constructive feedback. If you feel it is appropriate, we would greatly appreciate your reconsideration of the score.

---

### Official Review · Reviewer_NS41 · 2026-03-13

**Soundness:** 3
**Presentation:** 3
**Significance:** 3
**Originality:** 3
**Overall Recommendation:** 3
**Confidence:** 3

**Summary:**

This paper addresses the computational bottleneck of the Shampoo optimizer, specifically the high cost of frequent matrix inverse-root operations. While practitioners often use "stale" updates to save time, this heuristic typically lacks theoretical justification and can compromise training stability. The authors bridge this gap by establishing a theoretical link between staleness, operator error, and numerical instability.

The core contribution is FOAM (Frequency and Operator-error based Adaptive Damping Method), an adaptive framework that treats the damping factor $\epsilon$ as a dynamic control variable to suppress errors induced by staleness. FOAM utilizes a computationally efficient error proxy—derived from matrix perturbation theory—to sense when the current damping is insufficient and when a full eigendecomposition refresh is strictly necessary.

Experimental results on ViT and Conformer architectures demonstrate that FOAM significantly reduces the frequency of expensive eigendecompositions, often by over 80%, leading to substantial wall-clock speedups while maintaining robust convergence.

**Compliance With Llm Reviewing Policy:**

Affirmed.

**Key Questions For Authors:**

- Algorithm 3 uses max-based logic, while Section 6.1 describes a multiplicative rule. Which was used in experiments, and how does this affect sensitivity to learning rate scales?
- Calculating $h_t$ involves matrix multiplications. Can you provide a time breakdown: what is the % of time spent on the "sensing phase" per iteration, and does it scale to higher dimensions?
- Is the current task coverage sufficient to support the claims of broad generality? Additionally, when compared horizontally with recent preconditioning techniques mentioned, where exactly are the boundaries of FOAM's advantages?

**Limitations:**

yes

**Strengths And Weaknesses:**

## Strengths

- The paper precisely addresses the core bottleneck of the Shampoo optimizer in industrial applications—prohibitive computational overhead from matrix inversion—and provides a theoretical study for the common practice of "stale updates"
- It proposes the FOAM framework, which cleverly uses the damping factor as a "stability regulator," dynamically balancing optimization fidelity and computational efficiency via a lightweight error proxy.
- The paper establishes operator-gap bounds using matrix perturbation theory, clarifying the quantitative relationship between refresh frequency, damping levels, and training stability.

## Weaknesses

- The evaluation is limited to only two workloads and lacks direct comparisons with recent Shampoo variants or other stabilization techniques, making it hard to verify if FOAM is the superior solution.
- We noticed a discrepancy between the pseudocode and the text regarding the damping update logic. This inconsistency makes the implementation details unclear and may hinder the reproducibility of our results.
- The paper argues that FOAM saves time by reducing EVD frequency, and Figure 3 indeed shows fewer EVD calls. But the paper never reports the cost of computing the proxy or the total optimizer overhead decomposition. If the proxy is cheap, show it. If not, the wall-clock story is incomplete.

---

> ### Author Rebuttal · Authors · 2026-03-31
>
> We thank the reviewer for the constructive feedback, which highlights FOAM's ability to address Shampoo's inversion bottleneck.  We provide our responses below. We remain fully available for any follow-up discussions and will address any further inquiries promptly to ensure a clear evaluation.
>
> ---
>
> **1. Clarification regarding the update rule in Algorithm 3: $\epsilon_t=\max \left(\epsilon_0,\;\epsilon_{t-1}\frac{h_t}{\tau}\right).$**
>
> Our phrase *multiplicative feedback rule* in Section 6.1 was intended to describe this same floor-constrained multiplicative controller, not a different pure multiplicative rule. The role of the $\(\max(\epsilon_0,\cdot)\)$ term is simply to enforce a damping floor, preventing $\(\epsilon_t\)$ from shrinking below the baseline level where the inverse-root map becomes overly sensitive to perturbations. We will revise the wording to make clear that Section 6.1 and Algorithm 3 describe the same controller.
>
> ---
>
> **2. Overhead analysis**
>
> We measured proxy-computation time and EVD time separately, and reported each as a percentage of the total optimizer time over the full run.
>
> **2-1 Comparison of Computational Overhead Between Proxy Calculation and EVD**
>
> We profiled the optimizer's computational overhead and summarized the results below.
> | Model | Setting | Proxy computation cost | EVD cost |
> |---|---|---:|---:|
> | ViT | $(\tau=0.5,\; f=25,\; \epsilon_{\max}=3\times10^{-7})$ | $\sim$ 4.72% of total optimizer time | $\sim 17.67\%$ of total optimizer time |
> | ViT | $(\tau=0.1,\; f=50,\; \epsilon_{\max}=2\times10^{-7})$ | $\sim$ 2.10% of total optimizer time | $\sim 29.48\%$ of total optimizer time |
> | GPT-2 | $(\tau=0.25,\; f=40,\; \epsilon_{\max}=2\times10^{-7})$ | $\sim$ 3.8% of total optimizer time | $\sim 15.2\%$ of total optimizer time |
>
> **2-2 Increase in proxy  computational overhead by matrix size**
>
> We also recorded (Kronecker) factor-level timings by matrix dimension, which showed that, across ViT factor sizes (384-1024), proxy computation remained nearly flat and was much smaller than the EVD cost.
> | Model | Setting | Dimensions  | profiling |
> |---|---|---:|---:|
> | ViT | $(\tau=0.5,\; f=25,\; \epsilon_{\max}=3\times10^{-7}\)$ | 384 | 0.256ms |
> | ViT | same | 512 | 0.264ms |
> | ViT | same  | 1024 | 0.286ms |
> | GPT-2 |  $\(\tau=0.25,\; f=40,\; \epsilon_{\max}=2\times10^{-7}\)$ | 512 | 0.267ms |
> | GPT-2 |  same | 1024 | 0.285ms |
>
> Thus, when comparing the corresponding median eigendecomposition times were 13.29 ms, 18.44 ms, and 14.98 ms, the proxy remained roughly 50–70× cheaper than eigendecomposition across the real-factor dimensions encountered during training. In GPT-2, the proxy computation time likewise remained essentially flat across the dimensions 512 and 1024. Thus, proxy computation remains small relative to the EVD cost and does not meaningfully increase with factor dimension.
>
> ---
>
> **3. Generalization capability and boundaries of FOAM**
>
> **3.1 Additional Comparative Experiments Between FOAM and Shampoo-Based Methods**
>
>  We have added direct comparisons against the diagonalization-residual trigger baseline and SOAP.
> For the Eschenhagen-style baseline, we used the same diagonalization-residual trigger criterion, but refreshed with EVD rather than QR refinement. Our reason is that, in the trigger states observed in our runs, the amount of QR refinement needed to satisfy the practical residual criterion was already comparable to, or slower than, direct eigendecomposition. (Refer to the answer to reviewer AdF1) We will clarify this implementation choice explicitly so that the comparison is transparent.
>
> | Method | $\tau$ | $f$ | $\epsilon_{\max}$ | Wall-clock time (total: 60 epochs) | Validation accuracy |
> |---|---:|---:|---:|---:|---:|
> | FOAM | 0.1 | 50 | $(2\times10^{-7}\)$ | 1461 min | 74.05\% |
> | Eschenhagen et al. | 0.1 | 50 | N/A | 1486 min | 73.63\% |
> | SOAP | N/A | 50 | N/A | 1462 min | 74.00\% |
>
> To further evaluate FOAM on language models, we also conducted a pretraining experiment with GPT-2 (small) on the WikiText-103-v1 dataset.  Our training setup is as follows. Mini-batch size = 128, learning rate = 0.001, warm-up = 5% of total iterations followed by cosine decay, $(f=40\), \(\tau=0.25\), \(\epsilon_0 = 10^{-9}\)$, and $(\epsilon_{\max}=2\times10^{-7}, 3\times10^{-7}\)$.
>
> | Method | Wall-clock time| train loss |
> |---|---:|---:|
> | Stale Shampoo |  435min | 3.22 |
> | FOAM | 369 min | 3.20 |
> | FOAM | 366 min | 3.21 |
> | Eschenhagen et al. | 405 min | 3.20 |
> | SOAP | 348 min | 3.28 |
>
> **3.2 Boundaries of FOAM**
>
> First, there is an increase in tuning costs due to the introduction of two new hyperparameters $\tau, \epsilon_{\max}$. Second, although FOAM clearly performs fewer operations than Shampoo, it still carries out EVD, which is one of the bottlenecks in directly applying FOAM to LLM tasks. In other words, we believe the development of preconditioned methods that do not require EVD itself is an important area for future research.

---

> > ### Author Rebuttal · Reviewer_NS41 · 2026-04-04
> >
> > Thank you for the detailed rebuttal. The response helps clarify several of my concerns. In particular, the clarification of the update rule in Algorithm 3 resolves the ambiguity between Section 6.1 and the algorithmic description, and the newly provided overhead breakdown is useful. I also appreciate the additional comparisons against Eschenhagen-style methods and SOAP, as these experiments improve the empirical positioning of FOAM.
> >
> > That said, my concerns are only partially resolved. The rebuttal itself confirms that FOAM still relies on EVD, and the reported overhead suggests that EVD remains a nontrivial component of the optimizer time. Therefore, I still have some reservations about how strongly the paper’s efficiency and scalability claims transfer to larger-scale language model training. The added GPT-2 small result is helpful, but it is still somewhat limited evidence for broader LLM applicability.
> >
> > I would also appreciate clearer exposition in the final version regarding the comparison protocol for the Eschenhagen-style baseline, especially the choice to refresh with EVD rather than QR refinement, since this is important for interpreting the fairness of the comparison. Overall, the rebuttal improves clarity and strengthens the paper, but I view the scalability/generalization claims as only partially addressed at this stage.

---

> > > ### Author Response · Authors · 2026-04-06
> > >
> > > First, we are glad that our response has partially addressed your concerns. We also appreciate the opportunity to provide further clarification on your remaining concerns. Below are our responses to your remaining concerns.
> > >
> > > ---
> > >
> > > **1. EVD Component and Overhead**
> > >
> > > FOAM resolves the fundamental trade-off between performance and computational cost by significantly streamlining the EVD overhead, establishing itself as an efficient alternative to the non-diagonal preconditioner Shampoo. While Shampoo offers superior performance over AdamW, the diagonal preconditioner, they typically suffer from high computational cost. Our primary contribution lies in addressing this bottleneck; by utilizing an error proxy $h_t$, FOAM achieves lower computational cost with higher performance than Shampoo.
> > >
> > > To further clarify this, we provide a comparison of Conformer performance on LibriSpeech and overhead below:
> > > | Optimizer | Train Loss | Wall-clock Time |
> > > | :--- | :---: | :---: |
> > > | **Adam** |0.158 | 242 |
> > > | **Shampoo** | 0.145 | 257 |
> > > | **FOAM** | 0.120 | 255 |
> > >
> > > As shown in the table, FOAM demonstrates superior cost-efficiency by effectively minimizing the computational burden. Compared to the Adam baseline, FOAM achieves a 1.32x performance improvement (Loss: 0.158→0.12) with only a 5.4% increase in wall-clock time, a trade-off that is highly reasonable given the substantial accuracy gain. Most notably, FOAM significantly reduces the structural overhead associated with non-diagonal preconditioning. While the Shampoo requires an additional 15 minutes of computation over Adam, FOAM cuts this overhead by 13.3% while simultaneously delivering 1.21x higher performance than Shampoo. These results highlight FOAM’s ability to provide superior performance while narrowing the efficiency gap between diagonal and non-diagonal optimizers.
> > >
> > > ---
> > >
> > > **2. Applicability to Recent Language Models**
> > >
> > > To further alleviate concerns regarding applicability, we conducted additional experiments across diverse model architectures, scales, and GPU environments, including two sizes of Nano-GPT [2] and SmolLM2 [3] (over 70k monthly downloads on Hugging Face), and confirmed FOAM’s robust effectiveness and superior efficiency.
> > >
> > > - Hardware: 4 $\times$ H200  for NanoGPT and  8 $\times$ H100 for SmolLM2
> > > - Hyperparameters : $f$: 50, $\tau$: 0.25, $\epsilon_0$: $10^{-10}$, $\epsilon_{\max}$: $10^{-8}$
> > >
> > > | Model Family | Parameters | Optimizer | Mini-batch Size | Iterations | Train Loss | Wall-clock Time (minute) |
> > > | :--- | :--- | :--- | :--- | :--- | :--- | :--- |
> > > | NanoGPT | 125M | Stale Shampoo | 512 | 5,000 | 2.906 | 150 |
> > > | NanoGPT | 125M | FOAM | 512 | 5,000 | 2.869 | 137 |
> > > | NanoGPT | 225M | Stale Shampoo | 512 | 8,000 | 2.04 | 376 |
> > > | NanoGPT | 225M | FOAM | 512 | 8,000 | 1.99 | 374 |
> > > | SmolLM2 | 360M | Stale Shampoo | 512 | 5,000 | 2.32 | 189 |
> > > | SmolLM2 | 360M | FOAM | 512 | 5,000 | 2.23 | 181 |
> > >
> > >
> > > The results in the table above demonstrate that FOAM consistently delivers robust performance across various model architectures and scales, achieving lower training loss with reduced wall-clock time compared to Stale Shampoo (e.g., an 8.7% time reduction for NanoGPT-125M). These consistent gains on NanoGPT and SmolLM2 using H100/H200 GPUs indicate that FOAM is highly applicable and scalable to modern language models, effectively addressing the efficiency requirements of larger-scale training.
> > >
> > > [1] Benchmarking neural network training algorithms. arXiv:2306.07179 (2023).
> > >
> > > [2] https://github.com/karpathy/nanogpt?tab=readme-ov-file
> > >
> > > [3] SmolLM2: When Smol Goes ... arXiv:2502.02737 (2025).
> > >
> > > ---
> > >
> > > **3. About Implementation of baseline [4] in terms of QR and EVD**
> > > We appreciate the reviewer’s feedback regarding the comparison protocol for the baseline [4] . We realize that our previous statement:
> > > > “we used the same diagonalization…, but refreshed with EVD rather than QR refinement”
> > >
> > > may have inadvertently suggested an arbitrary modification on our part. We apologize for the lack of clarity and would like to provide the following correction to ensure a fair interpretation of our results.
> > >
> > >
> > > Our decision to use EVD instead of QR refinement was not an arbitrary substitution; rather, it was a deliberated choice to follow the most performant configuration identified by the original authors of [4]. Study. The authors noted that (second par. of Section 4.2 in [4]):
> > > > The QR algorithm was slightly more expensive than computing eigh. ...
> > >
> > > > The default configuration of SOAP, was also slightly slower in wall-clock time compared to adaptively computing eigh and reached a worse final loss.
> > >
> > >
> > > Guided by this, we implemented the baseline using the author-recommended EVD-based adaptive approach, as it represents the strongest version of their algorithm in terms of both loss and wall-clock time. Therefore, our comparison is conducted fairly, and we will dtail this protocol choice in the final version.
> > >
> > >
> > > [4] Purifying Shampoo: Investigating Shampoo’s ... Eschehagen et al.

---

### Official Review · Reviewer_AdF1 · 2026-03-17

**Soundness:** 3
**Presentation:** 3
**Significance:** 3
**Originality:** 2
**Overall Recommendation:** 4
**Confidence:** 2

**Summary:**

The authors provide a first formal theoretical connection between damping magnitude and error from staleness-induced operators. They propose an algorithm, FOAM, where they dynamically regulate the damping factor to enhance numerical stability. Algorithm is validated through experiments on large scale benchmarks.

**Compliance With Llm Reviewing Policy:**

Affirmed.

**Final Justification:**

The authors have addressed my concerns in the review. Only minor remains are the hyperparamters sensitivity discussion can be provided with rigorous empirical experiments to provide as much intuition as possible for practitioners.

**Key Questions For Authors:**

See Strengths and Weakness Section

**Limitations:**

See Strengths and Weakness Section

**Strengths And Weaknesses:**

Strengths

1. Clear and solid framework that displays motivation to focus on damping with solid theoretical contributions.
2. Strong experimental examples showing improvements in wall-clock time.
3. Convincing that the proxy outperforms the diagonalization residual baseline.

Weaknesses

1. Theoretical concern regarding the proof of Lemma C.6. After defining $X$, $P$ and $Q$, in line 905, the proof then continues with $X \preceq P$ and $X \preceq Q$, then states that $P$ and $Q$ commute, then concludes that their common eigenbasis one has $X \preceq \text{min}(P,Q) \preceq (PQ)^{1/2}$.

However, simultaneous diagonalization of $P$ and $Q$ do not make the Loewer inequalities $X \preceq P$ and $X \preceq Q$ unless $X$ also shares that eigenbasis, which is not established. We can construct PSD $X$ and diagonal PSD $P$,$Q$ such that $X \preceq P$ and $X \preceq Q$ but $\text{min}(P,Q) \npreceq (PQ)^{1/2}$

See this example below.

 Let
$P = \begin{pmatrix} 2 & 0 \\\\ 0 & 1 \end{pmatrix},
\qquad
Q = \begin{pmatrix} 1 & 0 \\\\ 0 & 2 \end{pmatrix},
\qquad
X = \begin{pmatrix} 0.9 & 0.3 \\\\ 0.3 & 0.9 \end{pmatrix}$,

We can verify that $P - X$ and $Q - X$ are both positive semidefinite
(both have determinant $0.02 > 0$ and positive trace), so
$X \preceq P$ and $X \preceq Q$.  However,
$$
  \min(P,Q) - X = I - X
  = \begin{pmatrix} 0.1 & -0.3 \\\\ -0.3 & 0.1 \end{pmatrix},
$$

which has determinant $0.01 - 0.09 = -0.08 < 0$ and is therefore
not PSD. Hence $X \not\preceq \min(P,Q)$

This lemma is used to prove Theorem 5.4 (The regret bound).

2. I am a bit confused in regards to picking $\varepsilon_{max}, \tau, f$. The authors argue that FOAM is robust across a range of these values. Can authors further emphasize more on their theoretical findings and intuition on picking the most practical hyperparameters? There seems to lack a general formula/framework for setting them up for a new problem without running a full sweep.

3. Authors theoretically argue in Appendix I that the diagonalization-residual trigger is inefficient because of "Cold start" QR iterations. Paper can be strengthened by having numerical experiments to quantitatively back this up.  Authors can report the actual average number of QR iterations triggered during practical the experiments.

4. Authors should consider including Eschenhagen et al's method as a direct empirical baseline in their main wall-clock experiments, in addition to comparing against fixed-frequency stale shampoo baseline

5. Regarding the claim on being the first to use damping to combat staples, consider discussing/citing this work https://arxiv.org/pdf/1802.07928, as I believe they analyze convergence degradation cased by stale gradients and then introduce dampening mechanism to rekindle convergence.

---

> ### Author Rebuttal · Authors · 2026-03-31
>
> We thank the reviewer for providing such constructive feedback and for noting the theoretical motivation of our damping-oriented framework, its wall-clock speedups, and its superiority over the diagonalization residual baseline. We welcome any further feedback if our responses require more detail or clarification.
>
> ---
>
> **1. Response to a Question Regarding the Proof of Lemma C.6**
> We have removed that step and replaced it with a corrected derivation based on the Kubo-Ando geometric mean
> $ A \circ B := A^{1/2}(A^{-1/2} B A^{-1/2})^{1/2} A^{1/2}.$
> Since $\epsilon > 0$, the matrices involved are positive definite. From $X \preceq P$ and $X \preceq Q$, monotonicity of the matrix geometric mean gives: $X = X \circ X \preceq P \circ Q.$
> Because $P$ and $Q$ commute in our setting,
> $ P \circ Q = P^{1/2}Q^{1/2}$, and by the Kronecker-product structure,
> $P^{1/2}Q^{1/2} = B_m^{1/2} \otimes A_n^{1/2}.$
> This gives the corrected Lemma C.6. We will update the appendix accordingly and make the proof fully self-contained.
>
> ---
>
> **2. On practical hyperparameter recommendations.**
> We appreciate the reviewer's comment for its clarity. As pointed out, the manuscript does not currently provide explicit guidelines for hyperparameter configuration. We intend to provide the following guidelines in the Appendix of the revised manuscript.
> Our intended interpretation is that these are control parameters with distinct roles: $f$ is the sensing/check period, so larger $f$ reduces overhead but increases stale drift; $\tau$ is the tolerated proxy level, so smaller $\tau$ makes FOAM react more aggressively by increasing damping earlier; and $\epsilon_{\max}$ is an over-damping cap, introduced because larger damping suppresses staleness-induced error but, if made too large, weakens the useful preconditioning effect. Based on this interpretation, our practical guideline is to start from an $f$ that is already stable for stale Shampoo on the task (or slightly smaller), then choose a moderate $\tau$ and an $\epsilon_{\max}$ large enough to absorb temporary drift but small enough to avoid over-damping. We will revise the paper to make this control-based interpretation and initialization procedure explicit.
>
> ---
>
> **3. On quantitatively supporting the Appendix I claim.**
> We have now quantitatively backed up our Appendix I claim using trigger states collected from practical residual-trigger runs.
>
> | Model | Practical residual threshold | QR iters | Median runtime / direct `eigh` | Median residual after QR |
> |---|---:|---:|---:|---:|
> | ViT | 0.10 | 1 | 0.52× | 0.695 |
> | ViT | 0.10 | 2 | 1.01× | 0.451 |
> | ViT | 0.10 | 4 | 1.97× | 0.230 |
> | ViT | 0.10 | 8 | 3.93× | 0.098 |
> | GPT-2 small | 0.25 | 1 | 0.69× | 0.783 |
> | GPT-2 small | 0.25 | 2 | 0.92× | 0.467 |
> | GPT-2 small | 0.25 | 4 | 1.75× | 0.202 |
> | GPT-2 small | 0.25 | 8 | 3.43× | 0.0886 |
>
> For ViT, the median residual first drops below the practical threshold (0.25) only at 4 QR iterations, so the median required QR iterations is 4; by then, the median runtime is already 1.97× that of direct EVD. For GPT-2, the median residual first drops below the threshold (0.25) at 4 QR iterations, so the median required QR iterations is 4; at that point, the median runtime is already 1.75× that of direct EVD. Thus, in the trigger states observed in practice, a small number of QR steps is typically too inaccurate to satisfy the residual criterion, whereas enough QR refinement to satisfy the criterion is already comparable to, or slower than, direct EVD. We will revise Appendix I to state this claim in this more quantitative form.
>
> ---
>
> **4. Comparison with Eschenhagen et al.**
> In the ViT experiment, we set both methods to $\tau = 0.1$ and $f = 50$, and additionally used $\epsilon_{\max} = 2\times10^{-7}$ for FOAM.
>
> | Method | $\tau$ | $f$ | $\epsilon_{\max}$ | Wall-clock time (60 epochs) | (full-batch) train loss | validation accuracy |
> |---|---:|---:|---:|---:|---:|---:|
> | Residual-trigger baseline | 0.1 | 50 | N/A | 1486 min | 0.60 | 73.63% |
> | FOAM | 0.1 | 50 | $2\times10^{-7}$ | 1461 min | 0.58 | 74.05% |
>
> Additionally, please refer to our response to **Question 3 from Reviewer NS41** for the GPT-2 experiments.
>
> Under this matched setup, FOAM achieved both lower wall-clock time and higher validation accuracy on ViT. In optimizer-time profiling, we also observed that the residual-trigger baseline performed more EVD than FOAM. While this contributes to the runtime difference, we believe the key mechanism is that FOAM adaptively adjusts $\epsilon_t$, whereas the residual-trigger baseline keeps damping fixed.
>
> ---
>
> **5. Further related work.**
> We agree that arXiv:1802.07928 is relevant at a high level and will add it to the Related Work section. We will state the following in the manuscript: Kardam dampens **stale gradients** in asynchronous SGD via a scalar weight, whereas our work controls **stale preconditioners** in Shampoo by adaptively tuning the inverse-root damping level $\epsilon$.

---

> > ### Author Rebuttal · Reviewer_AdF1 · 2026-04-03
> >
> > The reviewers have addressed my concerns. Only minor thing is for a method that adds more hyperparameters on top of Shampoo, it would be great if authors can further formalize, provide even greater detail and run exhaustive set of experiments showcasing parameter sensitivity. I will raise my score.

---

> > > ### Author Response · Authors · 2026-04-06
> > >
> > > We sincerely appreciate that our rebuttal has satisfactorily resolved your concerns.
> > >
> > > To further strengthen the paper, we will include additional experiments in the revised manuscript to provide more systematic evidence of FOAM’s robustness to the two additional hyperparameters. We will also clarify practical guidance for choosing these parameters so that the method is easier to apply in new settings.
> > >
> > > Thank you again for your thoughtful and constructive feedback.

---

### Decision · Program_Chairs · 2026-04-30

**Decision:**

Accept (regular)

**Comment:**

The reviewers generally found the submission idea useful, and the rebuttal resolved most important concerns, including the main proof issue, the damping-update clarification, and the empirical positioning against recent baselines.

The method does introduce additional hyperparameters, and the theory should be framed more carefully as a convex regret-style justification rather than a direct nonconvex convergence guarantee. There is also some added overhead from the method, but the rebuttal shows that this overhead is small relative to EVD and can be offset by improved performance and reduced wall-clock time in practice.

Overall, I view this as a technically solid and practically useful contribution, with limitations that are real but not disqualifying. I therefore recommend weak accept.